# Differentiable Equilibrium Computation with Decision Diagrams for Stackelberg Models of Combinatorial Congestion Games

**Shinsaku Sakaue**
The University of Tokyo
Tokyo, Japan
sakaue@mist.i.u-tokyo.ac.jp

**Kengo Nakamura**
NTT Communication Science Laboratories
Kyoto, Japan
kengo.nakamura.dx@hco.ntt.co.jp

## Abstract

We address Stackelberg models of combinatorial congestion games (CCGs); we aim to optimize the parameters of CCGs so that the selfish behavior of non-atomic players attains desirable equilibria. This model is essential for designing such social infrastructures as traffic and communication networks. Nevertheless, computational approaches to the model have not been thoroughly studied due to two difficulties: (I) bilevel-programming structures and (II) the combinatorial nature of CCGs. We tackle them by carefully combining (I) the idea of *differentiable* optimization and (II) data structures called *zero-suppressed binary decision diagrams* (ZDDs), which can compactly represent sets of combinatorial strategies. Our algorithm numerically approximates the equilibria of CCGs, which we can differentiate with respect to parameters of CCGs by automatic differentiation. With the resulting derivatives, we can apply gradient-based methods to Stackelberg models of CCGs. Our method is tailored to induce Nesterov's acceleration and can fully utilize the empirical compactness of ZDDs. These technical advantages enable us to deal with CCGs with a vast number of combinatorial strategies. Experiments on real-world network design instances demonstrate the practicality of our method.

## 1 Introduction

Congestion games (CGs) [49] form an important class of non-cooperative games and appear in various resource allocation scenarios. Combinatorial CGs (CCGs) can model more complex situations where each strategy is a combination of resources. A well-known example of a CCG is selfish routing [50], where each player on a traffic network chooses an origin-destination path, which is a strategy given by a combination of some roads with limited width (resources). Computing the outcomes of players' selfish behaviors (or equilibria) is essential when designing social infrastructures such as traffic networks. Therefore, how to compute equilibria of CGs has been widely studied [5, 12, 57, 42].

In this paper, we are interested in the perspective of the leader who designs non-atomic CCGs. For example, the leader aims to optimize some traffic-network parameters (e.g., road width values) so that players can spend less traveling time at equilibrium. An equilibrium of non-atomic CCGs is characterized by an optimum of potential function minimization [41, 53]. Thus, designing CCGs can be seen as a Stackelberg game; the leader optimizes the parameters of CCGs to minimize an objective function (typically, the social-cost function) while the follower, who represents the population of selfish non-atomic players, minimizes the potential function. This mathematical formulation is called the Stackelberg model in the context of traffic management [46]. Therefore, we call our model with general combinatorial strategies a Stackelberg model of CCGs.

35th Conference on Neural Information Processing Systems (NeurIPS 2021).

Stackelberg models of CCGs have been studied for cases where the potential function minimization has desirable properties. For example, Patriksson and Rockafellar [46] proposed a descent algorithm for traffic management using the fact that projections onto flow polyhedra can be done efficiently. However, many practical CCGs have more complicated structures. For example, in communication network design, each strategy is given by a Steiner tree (see [24, 42] and Section 1.1), and thus the projection (and even optimizing linear functions) is NP-hard. How to address such computationally challenging Stackelberg models of CCGs has not been well studied, despite its practical importance.

Inspired by a recent equilibrium computation method [42], we tackle the combinatorial nature of CCGs by representing their strategy sets with *zero-suppressed binary decision diagrams* (ZDDs) [39, 32], which are well-established data structures that provide empirically compact representations of combinatorial objects (e.g., Steiner trees and Hamiltonian paths). Although the previous method [42] can efficiently approximate equilibria with a Frank–Wolfe-style algorithm [18], its computation procedures break the differentiability of the outputs in the CCG parameters (the leader's variables), preventing us from obtaining gradient information required for optimizing leader's objective functions.

**Our contribution** is to develop a *differentiable* pipeline from leader's variables to equilibria of CCGs, thus enabling application of gradient-based methods to the Stackelberg models of CCGs. We smooth the Frank–Wolfe iterations using softmin, thereby making computed equilibria differentiable with respect to the leader's variables by automatic differentiation (or backpropagation). Although the idea of smoothing with softmin is prevalent [29, 38], our method has the following technical novelty:

- Our algorithm is tailored to induce Nesterov's acceleration, making both equilibrium computation and backpropagation more efficient. To the best of our knowledge, the idea of simultaneously making iterative optimization methods both differentiable and faster is new.
- Our method consists of simple arithmetic operations performed with ZDDs as in Algorithm 2. This is essential for making our equilibrium computation accept automatic differentiation. The per-iteration complexity of our method is linear in the ZDD size.

Armed with these advantages, our method can work with CCGs that have an enormous number of combinatorial strategies. We experimentally demonstrate its practical usefulness in real-world network design instances. Our method brings benefits by improving the designs of social infrastructures.

**Notation.** Let $[n] := \{1, \ldots, n\}$. For any $S \subseteq [n]$, $\mathbf{1}_S \in \{0, 1\}^n$ denotes a binary vector whose $i$-th entry is 1 if and only if $i \in S$. Let $\| \cdot \|$ be the $\ell_2$-norm.

## 1.1 Problem setting

We introduce the problem setting and some assumptions. For simplicity, we describe the *symmetric* setting, although our method can be extended to an *asymmetric* setting, as explained in Appendix A.

**Combinatorial congestion games (CCGs).** Suppose that there is an infinite amount of players with an infinitesimal mass (i.e., non-atomic). We assume the total mass is 1 without loss of generality. Let $[n]$ be a set of resources and let $\mathcal{S} \subseteq 2^{[n]}$ be a set of all feasible strategies. We define $d := |\mathcal{S}|$, which is generally exponential in $n$. Each player selects strategy $S \in \mathcal{S}$. Let $\boldsymbol{y} \in [0, 1]^n$ be a vector whose $i$-th entry indicates the total mass of players using resource $i \in [n]$. In other words, if we let $\boldsymbol{z} \in \triangle^d$ be a vector whose entry $z_S$ ($S \in \mathcal{S}$) indicates the total mass of players choosing $S$, we have $\boldsymbol{y} := \sum_{S \in \mathcal{S}} z_S \mathbf{1}_S \in \mathbb{R}^n$. Therefore, $\boldsymbol{y}$ is included in convex hull $\mathcal{C} := \left\{ \sum_{S \in \mathcal{S}} z_S \mathbf{1}_S \mid \boldsymbol{z} \in \triangle^d \right\}$, where $\triangle^d := \left\{ \boldsymbol{z} \in \mathbb{R}^d \mid \boldsymbol{z} \geq 0, \sum_{S \in \mathcal{S}} z_S = 1 \right\}$ is the $(d-1)$-dimensional probability simplex. A player choosing $S$ incurs cost $c_S(\boldsymbol{y}) := \sum_{i \in S} c_i(y_i)$, where each $c_i : \mathbb{R} \to \mathbb{R}$ is assumed to be strictly increasing; this corresponds to a natural situation where cost $c_i$ increases as $i \in [n]$ becomes more congested. Each player selfishly selects a strategy to minimize his/her own cost.

**Equilibrium and potential functions.** We say $\boldsymbol{z} \in \triangle^d$ attains a *(Wardrop) equilibrium* if for every $S \in \mathcal{S}$ such that $z_S > 0$, it holds that $c_S(\boldsymbol{y}) \leq \min_{S' \in \mathcal{S}} c_{S'}(\boldsymbol{y})$, where $\boldsymbol{y} := \sum_{S \in \mathcal{S}} z_S \mathbf{1}_S$. That is, no one has an incentive to deviate unilaterally. Let $f : \mathbb{R}^n \to \mathbb{R}$ be a *potential function* defined as $f(\boldsymbol{y}) := \sum_{i \in [n]} \int_0^{y_i} c_i(u) \mathrm{d}u$. From the first-order optimality condition, it holds that $\boldsymbol{z}$ attains an equilibrium iff $\boldsymbol{y} = \sum_{S \in \mathcal{S}} z_S \mathbf{1}_S$ satisfies $\boldsymbol{y} = \operatorname{argmin}_{\boldsymbol{u} \in \mathcal{C}} f(\boldsymbol{u})$. Note that minimizer $\boldsymbol{y}$ is unique since $c_i$ is strictly increasing, which means $f$ is strictly convex (not necessarily strongly convex).

**Stackelberg model of CCGs.** We turn to the problem of designing CCGs. For $i \in [n]$, let $c_i(y_i; \boldsymbol{\theta})$ be a cost function with parameters $\boldsymbol{\theta} \in \Theta$. We assume $c_i(y_i; \boldsymbol{\theta})$ to be strictly increasing in $y_i$ for any $\boldsymbol{\theta} \in \Theta$ and differentiable with respect to $\boldsymbol{\theta}$ for any $\boldsymbol{y} \in \mathcal{C}$. Let $f(\boldsymbol{y}; \boldsymbol{\theta}) = \sum_{i \in [n]} \int_0^{y_i} c_i(u; \boldsymbol{\theta}) \mathrm{d}u$ be a parameterized potential function, which is strictly convex in $\boldsymbol{y}$ for any $\boldsymbol{\theta} \in \Theta$. A leader who designs CCGs aims to optimize $\boldsymbol{\theta}$ values so that an objective function, $F : \Theta \times \mathcal{C} \to \mathbb{R}$, is minimized at an equilibrium of CCGs. Typically, $F$ is a social-cost function defined as $F(\boldsymbol{\theta}, \boldsymbol{y}) = \sum_{i \in [n]} c_i(y_i; \boldsymbol{\theta}) y_i$, which represents the total cost incurred by all players. Since an equilibrium is characterized as a minimizer of potential function $f$, the leader's problem can be written as follows:

$$\underset{\boldsymbol{\theta} \in \Theta}{\text{minimize}} \quad F(\boldsymbol{\theta}, \boldsymbol{y}) \qquad \text{subject to} \quad \boldsymbol{y} = \underset{\boldsymbol{u} \in \mathcal{C}}{\operatorname{argmin}} f(\boldsymbol{u}; \boldsymbol{\theta}). \tag{1}$$

Since minimizer $\boldsymbol{y}(\boldsymbol{\theta}) := \boldsymbol{y}$ is unique, we can regard $F(\boldsymbol{\theta}, \boldsymbol{y}(\boldsymbol{\theta}))$ as a function of $\boldsymbol{\theta}$. We study how to approximate derivatives of $F(\boldsymbol{\theta}, \boldsymbol{y}(\boldsymbol{\theta}))$ with respect to $\boldsymbol{\theta}$ for applying gradient-based methods to (1).

**Example 1: traffic management.** We are given a network with an origin-destination (OD) pair. Let $[n]$ be the edge set and let $\mathcal{S} \subseteq 2^{[n]}$ be the set of all OD paths. Each edge in the network has cost function $c_i(y_i; \boldsymbol{\theta})$, where $\boldsymbol{\theta}$ controls the width of the roads (edges). A natural example of the cost functions is $c(y_i; \boldsymbol{\theta}) = y_i / \theta_i$ for $\theta_i > 0$ (see, e.g., [46]), which satisfies the above assumptions, i.e., strictly increasing in $y_i$ and differentiable in $\theta_i$. Once $\boldsymbol{\theta}$ is fixed, players selfishly choose OD paths and consequently reach an equilibrium. The leader wants to find $\boldsymbol{\theta}$ that minimizes social cost $F$ at equilibrium, which can be formulated as a Stackelberg model of form (1). Note that although the Stackelberg model of standard selfish routing is well studied [46], there are various variants (e.g., routing with budget constraints [28, 42]) for which existing methods do not work efficiently.

**Example 2: communication network design.** We consider a situation where multi-site meetings are held on a communication network (see, e.g., [24, 42]). Given an undirected network with edge set $[n]$ and some vertices called terminals, groups of people at terminals hold multi-site meetings, including people at all the terminals. Since each group wants to minimize the communication delays caused by congestion, each selfishly chooses a way to connect all the terminals, namely, a Steiner tree covering all the terminals. If we let $c_i(y_i; \boldsymbol{\theta})$ indicate the delay of the $i$-th edge, a group choosing Steiner tree $S \in \mathcal{S}$ incurs cost $c_S(\boldsymbol{y}; \boldsymbol{\theta})$. As with the above traffic-management example, the problem of optimizing $\boldsymbol{\theta}$ to minimize the total delay at equilibrium can be written as (1).

## 1.2 Related work

Problems of form (1) arise in many fields, e.g., Stackelberg games [55], mathematical programming with equilibrium constraints [36], and bilevel programming [10, 13], which have been gaining attention in machine learning [17, 15]. Optimization problems with bilevel structures are NP-hard in most cases [22] (tractable cases include, e.g., when follower's problems are unconstrained and strongly convex [19], which does not hold in our case). Thus, how to apply gradient-based methods occupies central interest [15, 20]. In our Stackelberg model of CCGs, in addition to the bilevel structure, the follower's problem is defined on combinatorial strategy sets $\mathcal{S}$, further complicating it. Therefore, unlike the above studies, we focus on how to address such difficult problems by leveraging computational tools, including ZDDs [39] and automatic differentiation [34, 21].

Stackelberg models often arise in traffic management. Although many existing studies [46, 35, 6, 9] analyze theoretical aspects utilizing instance-specific structures (e.g., compact representations of flow polyhedra), applications to other types of realistic CCGs remain unexplored. By contrast, as with the previous method [42], our method is built on versatile ZDD representations of strategy sets, and the derivatives with respect to CCG parameters can be automatically computed with backpropagation. Thus, compared to the methods studied in traffic management, ours can be easily applied to and works efficiently with a broad class of realistic CCGs with complicated combinatorial strategies.

Our method is inspired by the emerging line of work on differentiable optimization [4, 60, 2, 51]. For differentiating outputs with respect to the parameters of optimization problems, two major approaches have been studied [20]: *implicit* and *iterative* differentiation. The first approach applies the implicit function theorem to equation systems derived from the Karush–Kuhn–Tucker (KKT) condition (akin to the single-level reformulation approach to bilevel programming). In our case, this approach is too expensive since the combinatorial nature of CCGs generally makes the KKT equation system exponentially large [16]. Our method is categorized into the second approach, which computes

a numerical approximation of an optimum with iterations of differentiable steps. This idea has yielded success in many fields [14, 37, 45, 7, 17]. Concerning combinatorial optimization, although differentiable methods for linear objectives are well studied [38, 60, 47, 8], no differentiable method has been developed for convex minimization on polytopes of, e.g., Steiner trees or Hamiltonian paths; this is what we need for dealing with the potential function minimization of CCGs. To this end, we use a Frank–Wolfe-style algorithm and ZDD representations of combinatorial objects.

## 2 Differentiable iterative equilibrium computation

We consider applying gradient-based methods (e.g., projected gradient descent) to problem (1). To this end, we need to compute the following gradient with respect to $\boldsymbol{\theta} \in \Theta \subseteq \mathbb{R}^k$ in each iteration:

$$\nabla F(\boldsymbol{\theta}, \boldsymbol{y}(\boldsymbol{\theta})) = \nabla_{\boldsymbol{\theta}} F(\boldsymbol{\theta}, \boldsymbol{y}(\boldsymbol{\theta})) + \nabla \boldsymbol{y}(\boldsymbol{\theta})^{\top} \nabla_{\boldsymbol{y}} F(\boldsymbol{\theta}, \boldsymbol{y}(\boldsymbol{\theta})),$$

where $\nabla_{\boldsymbol{\theta}} F(\boldsymbol{\theta}, \boldsymbol{y}(\boldsymbol{\theta}))$ and $\nabla_{\boldsymbol{y}} F(\boldsymbol{\theta}, \boldsymbol{y}(\boldsymbol{\theta}))$ denote the gradients with respect to the first and second arguments, respectively, and $\nabla \boldsymbol{y}(\boldsymbol{\theta})$ is the $n \times k$ Jacobian matrix.[1] The computation of $\nabla \boldsymbol{y}(\boldsymbol{\theta})$ is the most challenging part and requires differentiating $\boldsymbol{y}(\boldsymbol{\theta}) = \operatorname{argmin}_{\boldsymbol{u} \in \mathcal{C}} f(\boldsymbol{u}; \boldsymbol{\theta})$ with respect to $\boldsymbol{\theta}$. We employ the iterative differentiation approach for efficiently approximating $\nabla \boldsymbol{y}(\boldsymbol{\theta})$.

### 2.1 Technical overview

For computing equilibrium $\boldsymbol{y}(\boldsymbol{\theta})$, Nakamura et al. [42] solved potential function minimization with a variant of the Frank–Wolfe algorithm [33], whose iterations can be performed efficiently by using compact ZDD representations of combinatorial strategies. To the best of our knowledge, no other equilibrium computation methods can deal with various CCGs that have complicated combinatorial strategies, e.g., Steiner trees. Hence we build on [42] and extend their method to Stackelberg models.

First, we review the standard Frank–Wolfe algorithm. Starting from $\boldsymbol{x}_0 \in \mathcal{C}$, it alternately computes $\boldsymbol{s}_t = \operatorname{argmin}_{\boldsymbol{s} \in \mathcal{C}} \langle \nabla f(\boldsymbol{x}_t; \boldsymbol{\theta}), \boldsymbol{s} \rangle$ and $\boldsymbol{x}_{t+1} = (1 - \gamma_t)\boldsymbol{x}_t + \gamma_t \boldsymbol{s}_t$, where $\gamma_t$ is conventionally set to $\frac{2}{t+2}$. As shown in [18, 27], $\boldsymbol{x}_T$ has an objective error of $\mathrm{O}(1/T)$. Thus, we can obtain numerical approximation $\boldsymbol{y}_T(\boldsymbol{\theta}) = \boldsymbol{x}_T$ of equilibrium $\boldsymbol{y}(\boldsymbol{\theta})$ such that $f(\boldsymbol{y}_T(\boldsymbol{\theta}); \boldsymbol{\theta}) - f(\boldsymbol{y}(\boldsymbol{\theta}); \boldsymbol{\theta}) \leq \mathrm{O}(1/T)$. For obtaining gradient $\nabla \boldsymbol{y}_T(\boldsymbol{\theta})$, however, the above Frank–Wolfe algorithm does not work (neither does its faster variant used in [42]). This is because $\boldsymbol{s}_t = \operatorname{argmin}_{\boldsymbol{s} \in \mathcal{C}} \langle \nabla f(\boldsymbol{x}_t; \boldsymbol{\theta}), \boldsymbol{s} \rangle$ is piecewise constant in $\boldsymbol{\theta}$, which makes $\nabla \boldsymbol{y}_T(\boldsymbol{\theta})$ zero almost everywhere and undefined at some $\boldsymbol{\theta}$.

To resolve this issue, we develop a differentiable Frank–Wolfe algorithm by using softmin. We denote the softmin operation by $\boldsymbol{\mu}_{\mathcal{S}}(\boldsymbol{c})$ (detailed below). Since softmin can be seen as a differentiable proxy for argmin, one may simply replace $\boldsymbol{s}_t = \operatorname{argmin}_{\boldsymbol{s} \in \mathcal{C}} \langle \nabla f(\boldsymbol{x}_t; \boldsymbol{\theta}), \boldsymbol{s} \rangle$ with $\boldsymbol{s}_t = \boldsymbol{\mu}_{\mathcal{S}}(\eta_t \nabla f(\boldsymbol{x}_t; \boldsymbol{\theta}))$, where $\eta_t > 0$ is a scaling factor. Actually, the modified algorithm yields an $\mathrm{O}(1/T)$ convergence by setting $\eta_t = \Omega(t)$ (see [27, Theorem 1]). This modification, however, often degrades the empirical convergence of the Frank–Wolfe algorithm, as demonstrated in Section 4.1. We, therefore, consider leveraging softmin for acceleration while keeping the iterations differentiable. Based on an accelerated Frank–Wolfe algorithm [59], we compute $\boldsymbol{y}_T(\boldsymbol{\theta})$ as in Algorithm 1, and obtain $\nabla \boldsymbol{y}_T(\boldsymbol{\theta})$ by applying backpropagation. Furthermore, in Section 3, we explain how to efficiently compute $\boldsymbol{\mu}_{\mathcal{S}}(\boldsymbol{c})$ by using a ZDD-based technique [52]; importantly, its computation procedure also accepts the backpropagation.

While our work is built on the existing methods [59, 52, 42], none of them are intended to develop differentiable methods. A conceptual novelty of our work is its careful combination of those methods for developing a differentiable and accelerated optimization method, with which we can compute $\nabla \boldsymbol{y}_T(\boldsymbol{\theta})$. This enables the application of gradient-based methods to the Stackelberg models of CCGs.

### 2.2 Details of Algorithm 1

We compute $\boldsymbol{y}_T(\boldsymbol{\theta})$ with Algorithm 1. Note that $\boldsymbol{s}_t$, $\boldsymbol{c}_t$, and $\boldsymbol{x}_t$ depend on $\boldsymbol{\theta}$, which is not explicitly indicated for simplicity. The most crucial part is Step 5, where we use softmin rather than argmin to

---

[1]Although the derivatives of $\boldsymbol{y}(\boldsymbol{\theta})$ may not be unique, we abuse the notation and write $\nabla \boldsymbol{y}(\boldsymbol{\theta})$ for simplicity. As we will see shortly, we numerically approximate $\boldsymbol{y}(\boldsymbol{\theta})$ with $\boldsymbol{y}_T(\boldsymbol{\theta})$ whose derivative, $\nabla \boldsymbol{y}_T(\boldsymbol{\theta})$, exists uniquely. Therefore, when discussing our iterative differentiation method, we can ignore the abuse of notation.

---

**Algorithm 1** Differentiable Frank–Wolfe-based equilibrium computation

---

1: $\boldsymbol{s}_0 = \boldsymbol{c}_0 = \boldsymbol{0}$, $\boldsymbol{x}_{-1} = \boldsymbol{x}_0 = \boldsymbol{\mu}_{\mathcal{S}}(\boldsymbol{c}_0)$, and $\alpha_t = t$ $(t = 0, \ldots, T)$
2: **for** $t = 1, \ldots, T$ **:**
3:     $\boldsymbol{s}_t = \boldsymbol{s}_{t-1} - \alpha_{t-1}\boldsymbol{x}_{t-2} + (\alpha_{t-1} + \alpha_t)\boldsymbol{x}_{t-1}$
4:     $\boldsymbol{c}_t = \boldsymbol{c}_{t-1} + \eta\alpha_t\nabla f\left(\frac{2}{t(t+1)}\boldsymbol{s}_t;\boldsymbol{\theta}\right)$     ▷ $\nabla f(\boldsymbol{y};\boldsymbol{\theta})_i = c_i(y_i;\boldsymbol{\theta})$ is differentiable in $\boldsymbol{\theta}$
5:     Compute $\boldsymbol{x}_t = \boldsymbol{\mu}_{\mathcal{S}}(\boldsymbol{c}_t)$ with Algorithm 2     ▷ Differentiable softmin computation
     **return** $\boldsymbol{y}_T(\boldsymbol{\theta}) = \frac{2}{T(T+1)}\sum_{t=1}^{T}\alpha_t\boldsymbol{x}_t$

---

make the output differentiable in $\boldsymbol{\theta}$. Specifically, given any $\boldsymbol{c} \in \mathbb{R}^n$, we compute $\boldsymbol{\mu}_{\mathcal{S}}(\boldsymbol{c})$ as follows:

$$\boldsymbol{\mu}_{\mathcal{S}}(\boldsymbol{c}) \coloneqq \sum_{S \in \mathcal{S}} \mathbf{1}_S \frac{\exp\left(-\boldsymbol{c}^\top\mathbf{1}_S\right)}{\sum_{S' \in \mathcal{S}} \exp\left(-\boldsymbol{c}^\top\mathbf{1}_{S'}\right)}.$$

Intuitively, each entry in $\boldsymbol{c}$ represents the cost of each $i \in [n]$, and $\boldsymbol{c}^\top\mathbf{1}_S$ represents the cost of $S \in \mathcal{S}$. We consider a probability distribution over $\mathcal{S}$ defined by softmin with respect to costs $\{\boldsymbol{c}^\top\mathbf{1}_S\}_{S \in \mathcal{S}}$, and then marginalize it. The resulting vector is a convex combination of $\{\mathbf{1}_S\}_{S \in \mathcal{S}}$ and thus always included in $\mathcal{C}$. In the context of graphical modeling, this operation is called marginal inference [58]. Here, $\boldsymbol{\mu}_{\mathcal{S}}$ is defined by a summation over $\mathcal{S}$, and explicitly computing it is prohibitively expensive. Section 3 details how to efficiently compute $\boldsymbol{\mu}_{\mathcal{S}}$ by leveraging the ZDD representations of $\mathcal{S}$.

## 2.3 Convergence guarantee

Algorithm 1 is designed to induce Nesterov's acceleration [43, 59] and achieves an $\mathrm{O}(1/T^2)$ convergence, which is faster than the $\mathrm{O}(1/T)$ convergence of the original Frank–Wolfe algorithm.

**Theorem 1.** *Fix $\boldsymbol{\theta} \in \Theta$ and assume $f(\cdot;\boldsymbol{\theta})$ to be $L$-smooth on $\mathbb{R}^d$, i.e., $\Phi(\boldsymbol{z}) \coloneqq f(\sum_{S \in \mathcal{S}} z_S\mathbf{1}_S;\boldsymbol{\theta})$ $(\forall \boldsymbol{z} \in \mathbb{R}^d)$ satisfies $\Phi(\boldsymbol{z}') \leq \Phi(\boldsymbol{z}) + \langle\nabla\Phi(\boldsymbol{z}), \boldsymbol{z}' - \boldsymbol{z}\rangle + \frac{L}{2}\|\boldsymbol{z}' - \boldsymbol{z}\|^2$ for all $\boldsymbol{z}, \boldsymbol{z}' \in \mathbb{R}^d$.[2] If we let $\eta \in [\frac{1}{CL}, \frac{1}{4L}]$ for some $C > 4$, Algorithm 1 returns $\boldsymbol{y}_T(\boldsymbol{\theta})$ such that*

$$f(\boldsymbol{y}_T(\boldsymbol{\theta});\boldsymbol{\theta}) - f(\boldsymbol{y}(\boldsymbol{\theta});\boldsymbol{\theta}) \leq \mathrm{O}\left(\frac{CL\ln d}{T^2}\right).$$

We present the proof in Appendix B. In essence, softmin can be seen as a dual mirror descent step with the Kullback–Leibler divergence, and combining it with a primal gradient descent step yields the acceleration [3]. Although the acceleration technique itself is well studied, it has not been explored in the context of differentiable optimization. To the best of our knowledge, simultaneously making iterative optimization methods both differentiable and faster is a novel idea. This observation can be beneficial for developing other fast differentiable iterative algorithms. Experiments in Section 4.1 demonstrate that the acceleration indeed enhances the convergence speed in practice.

Note that the faster convergence enables us to more efficiently compute both $\boldsymbol{y}_T(\boldsymbol{\theta})$ and $\nabla\boldsymbol{y}_T(\boldsymbol{\theta})$. The latter is because Algorithm 1 with a smaller $T$ generates a smaller computation graph, which determines the computation complexity of the backpropagation for obtaining $\nabla\boldsymbol{y}_T(\boldsymbol{\theta})$. Therefore, Algorithm 1 is suitable as an efficient differentiable pipeline between $\boldsymbol{\theta}$ and $\boldsymbol{y}_T(\boldsymbol{\theta})$.

## 2.4 Implementation consideration: how to choose $\eta$ and $T$

While Theorem 1 suggests setting $\eta$ to $\frac{1}{4L}$ or less, this choice is often too conservative in practice. Thus, we should search for $\eta$ values that bring high empirical performances. When using Algorithm 1 as a subroutine of gradient-based methods, it is repeatedly called to solve similar equilibrium computation instances. Therefore, a simple and effective way for locating good $\eta$ values is to apply Algorithm 1 with various $\eta$ values to example instances, as we will do in Section 4.1. We expect the empirical performance to improve with a line search of $\eta$, which we leave for future work.

---

[2]Smoothness parameter $L$ defined on $\mathbb{R}^d$ can be, in general, exponentially large in $n$, albeit constant in $T$. How to alleviate the dependence on $L$ remains an open problem.

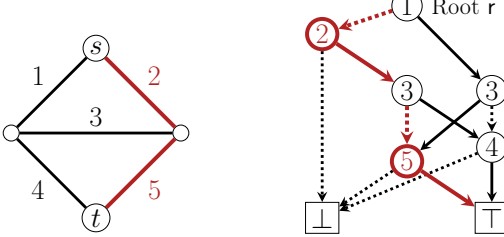

Figure 1: Example of ZDD $Z_{\mathcal{S}}$ (right), where $n = 5$ and $\mathcal{S}$ is a family of all simple $s$–$t$ paths (left). ZDD has two terminal nodes ($\top$ and $\bot$) and non-terminal nodes labeled by $l_v \in [n]$. Solid (dashed) arcs represent 1-arcs (0-arcs).

**Algorithm 2** Computation of $\boldsymbol{\mu}_{\mathcal{S}}(\boldsymbol{c})$ with ZDD $Z_{\mathcal{S}} = (V, A)$

1: $B_{\top} = 1$ and $B_{\bot} = 0$
2: **for** $v \in V \setminus \{\top, \bot\}$ (bottom-up) **:**
3: $\quad B_v = B_{c_v^0} + \exp(-c_{l_v}) \times B_{c_v^1}$
4: $P_r = 1$ and $P_v = 0$ ($v \in V \setminus \{r\}$)
5: $\boldsymbol{x} = (0, \ldots, 0)^{\top}$
6: **for** $v \in V \setminus \{\top, \bot\}$ (top-down) **:**
7: $\quad p^0 = B_{c_v^0}/B_v$ and $p^1 = 1 - p^0$
8: $\quad P_{c_v^0} \mathrel{+}= p^0 P_v$ and $P_{c_v^1} \mathrel{+}= p^1 P_v$
9: $\quad x_{l_v} \mathrel{+}= p^1 P_v$
10: **return** $\boldsymbol{x} = (x_1, \ldots, x_n)^{\top}$

To check whether the Frank–Wolfe algorithm has converged or not, we usually use the Frank–Wolfe gap [27], an upper-bound on an objective error. To obtain high-quality solutions, we terminate the algorithm when the gap becomes sufficiently small. In our case, however, our purpose is to obtain gradient information $\nabla \boldsymbol{y}_T(\boldsymbol{\theta})$, and $\boldsymbol{y}_T(\boldsymbol{\theta})$ with a small $T$ sometimes suffices to serve the purpose. By using a small $T$, we can reduce the computation complexity. Experiments in Section 4.2 show that the performance of a projected gradient method that uses $\nabla \boldsymbol{y}_T(\boldsymbol{\theta})$ is not so sensitive to $T$ values.

## 3 Efficient softmin computation with decision diagrams

This section describes how to efficiently compute $\boldsymbol{\mu}_{\mathcal{S}}(\boldsymbol{c})$ by leveraging ZDD representations of $\mathcal{S}$. The main idea is to apply a technique called weight pushing [40] (or path kernel [56]) to ZDDs. A similar idea was used for obtaining efficient combinatorial bandit algorithms [52], but this research does not use it to develop differentiable algorithms. Our use of weight pushing comes from another important observation: it consists of simple arithmetic operations that accept reverse-mode automatic differentiation with respect to $\boldsymbol{c}$, as shown in Algorithm 2. In other words, Algorithm 2 does not use, e.g., $|\cdot|$ or argmin. Therefore, ZDD-based weight pushing can be incorporated into the pipeline from $\boldsymbol{\theta}$ to $\boldsymbol{y}_T(\boldsymbol{\theta})$ without breaking the differentiability.

### 3.1 Zero-suppressed binary decision diagrams

Given set family $\mathcal{S} \subseteq 2^{[n]}$, we explain how to represent it with ZDD $Z_{\mathcal{S}} = (V, A)$, a DAG-shaped data structure (see, e.g., Figure 1). The node set, $V$, has two terminal nodes $\top$ and $\bot$ (they represent true and false, respectively) and non-terminal nodes. There is a single root, $r \in V \setminus \{\bot, \top\}$. Each $v \in V \setminus \{\bot, \top\}$ has label $l_v \in [n]$ and two outgoing arcs, 1- and 0-arcs, which indicate whether $l_v$ is chosen or not, respectively. Let $c_v^0, c_v^1 \in V$ denote two nodes pointed by 0- and 1-arcs, respectively, outgoing from $v$. For any $v \in V \setminus \{\top, \bot\}$, let $\mathcal{R}_v \subseteq 2^A$ be the set of all directed paths from $v$ to $\top$. We define $\mathcal{R} := \bigcup_{v \in V \setminus \{\top, \bot\}} \mathcal{R}_v$. For any $R \in \mathcal{R}$, let $X(R) := \{ l_v \in [n] \mid (v, c_v^1) \in R \}$, i.e., labels of tails of 1-arcs belonging to R. ZDD $Z_{\mathcal{S}}$ represents $\mathcal{S}$ as a set of $r$–$\top$ paths: $\mathcal{S} = \{ X(R) \mid R \in \mathcal{R}_r \}$. There is a one-to-one correspondence between $S \in \mathcal{S}$ and $R \in \mathcal{R}_r$, i.e., $S = X(R)$.

Figure 1 presents an example of ZDD $Z_{\mathcal{S}}$, where $\mathcal{S}$ is the family of simple $s$–$t$ paths. For example, $S = \{2, 5\} \in \mathcal{S}$ is represented in $Z_{\mathcal{S}}$ by the red path, $R \in \mathcal{R}_r$, with labels $\{1, 2, 3, 5, \top\}$. The labels of the tails of the 1-arcs form $X(R) = \{2, 5\}$, which equals $S$.

We define the size of $Z_{\mathcal{S}} = (V, A)$ by $|Z_{\mathcal{S}}| := |V|$. Note that $|A| \leq 2 \times |Z_{\mathcal{S}}|$ always holds. In general, the ZDD sizes and the complexity of constructing ZDDs can be exponential in $n$. Fortunately, many existing studies provide efficient methods for constructing compact ZDDs. One such method is the frontier-based search [30], which is based on Knuth's Simpath algorithm [32]. Their method is particularly effective when $\mathcal{S}$ is a family of network substructures such as Hamiltonian paths, Steiner trees, matchings, and cliques. Furthermore, the family algebra [39, 32] of ZDDs enables us to deal with various logical constraints. Using those methods, we can flexibly construct ZDDs for various complicated combinatorial structures, e.g., Steiner trees whose size is at most a certain value.

Moreover, we can sometimes theoretically bound the ZDD sizes and the construction complexity. For example, if $\mathcal{S}$ consists of the aforementioned substructures on network $G = (V, E)$ with a constant pathwidth, the ZDD sizes and the construction complexity are polynomial in $|E|$ [30, 26].

### 3.2 Details of Algorithm 2 and computation complexity

Algorithm 2 computes $\boldsymbol{x} = \boldsymbol{\mu}_{\mathcal{S}}(\boldsymbol{c})$ for any $\boldsymbol{c} = (c_1, \ldots, c_n)^\top \in \mathbb{R}^n$. First, it computes $\{B_v\}_{v \in V}$ in a bottom-up topological order of $Z_{\mathcal{S}}$. Note that $B_v = \sum_{S \in \{X(R) \mid R \in \mathcal{R}_v\}} \exp(-\sum_{i \in S} c_i)$ holds. Then it computes $\{P_v\}_{v \in V}$. Each $P_v$ indicates the probability that a top-down random walk starting from root node r reaches $v \in V$, where we choose 0-arc (1-arc) with probability $p^0$ ($p^1$). From $B_\bot = 0$ and the construction of $\{B_v\}_{v \in V}$, the random walk never reaches $\bot$, and its trajectory $R \in \mathcal{R}_r$ recovers $X(R) \in \mathcal{S}$ with a probability proportional to $\exp(-\sum_{i \in X(R)} c_i) = \exp(-\boldsymbol{c}^\top \mathbf{1}_{X(R)})$. Therefore, by summing the probabilities of reaching $v$ and choosing a 1-arc outgoing from $v$ for each $i \in [n]$ as in Step 9, we obtain $\boldsymbol{x} = \boldsymbol{\mu}_{\mathcal{S}}(\boldsymbol{c})$. In practice, we recommend implementing Algorithm 2 with the log-sum-exp technique and double-precision computations for numerical stability.

Algorithm 2 runs in $O(|Z_{\mathcal{S}}|)$ time, and thus Algorithm 1 takes $O((n + C_\nabla + |Z_{\mathcal{S}}|)T)$ time, where $C_\nabla$ is the cost of computing $\nabla f$. From the cheap gradient principle [21], the complexity of computing $\nabla F(\boldsymbol{\theta}, \boldsymbol{y}_T(\boldsymbol{\theta}))$ with backpropagation is almost the same as that of computing $F(\boldsymbol{\theta}, \boldsymbol{y}_T(\boldsymbol{\theta}))$. That is, smaller ZDDs make the computation of both $\boldsymbol{y}_T(\boldsymbol{\theta})$ and $\nabla \boldsymbol{y}_T(\boldsymbol{\theta})$ faster. Therefore, our method significantly benefits from the empirical compactness of ZDDs. Note that we can construct ZDD $Z_{\mathcal{S}}$ in a preprocessing step; once we obtain $Z_{\mathcal{S}}$, we can reuse it every time $\boldsymbol{\mu}_{\mathcal{S}}(\cdot)$ is called.

## 4 Experiments

Section 4.1 confirms the benefit of acceleration to empirical convergence speed. Section 4.2 demonstrates the usefulness of our method via experiments on communication network design instances. Section 4.3 presents experiments with small instances to see whether our method can empirically find globally optimal $\boldsymbol{\theta}$. Due to space limitations, we present full experimental results in Appendix C.

All the experiments were performed using a single thread on a 64-bit macOS machine with 2.5 GHz Intel Core i7 CPUs and 16 GB RAM. We used C++11 language, and the programs were compiled by Apple clang 12.0.0 with `-O3 -DNDEBUG` option. We used Adept 2.0.5 [23] as an automatic differentiation package and Graphillion 1.4 [25] for constructing ZDDs, where we used a beam-search-based path-width optimization method [26] to specify the traversal order of edges. The source code is available at https://github.com/nttcslab/diff-eq-comput-zdd.

**Problem setting.** We address Stackelberg models for optimizing network parameters $\boldsymbol{\theta} \in \mathbb{R}^n$, where $[n]$ represents an edge set. We focus on two situations where combinatorial strategies $\mathcal{S} \subseteq 2^{[n]}$ are Hamiltonian cycles and Steiner trees. The former is a variant of the selfish-routing setting, and the latter arises when designing communication networks as in Section 1.1. Note that in both settings, common operations on $\mathcal{C}$, e.g., projection and linear optimization, are NP-hard. We use two types of cost functions: fractional cost $c_i(y_i; \boldsymbol{\theta}) = d_i(1 + C \times y_i/(\theta_i + 1))$ and exponential cost $c_i(y_i; \boldsymbol{\theta}) = d_i(1 + C \times y_i \exp(-\theta_i))$, where $d_i \in (0, 1]$ is the length of the $i$-th edge (normalized so that $\max_{i \in [n]} d_i = 1$ holds) and $C > 0$ controls how heavily the growth in $y_i$ (congestion) affects cost $c_i$. We set $C = 10$. Note that edge $i$ with a larger $\theta_i$ is more tolerant to congestion. The leader aims to minimize social cost $F(\boldsymbol{\theta}, \boldsymbol{y}(\boldsymbol{\theta})) = \sum_{i \in [n]} c_i(y_i(\boldsymbol{\theta}); \boldsymbol{\theta})y_i(\boldsymbol{\theta})$. In realistic situations, the leader cannot let all edges have sufficient capacity due to budget constraints. To model this situation, we impose a constraint on $\boldsymbol{\theta}$ by defining $\Theta = \{\boldsymbol{\theta} \in \mathbb{R}^n_{\geq 0} \mid \boldsymbol{\theta}^\top \mathbf{1} = n\}$, where $\mathbf{1}$ is the all-one vector.

**Datasets.** Table 1 summarizes the information about datasets and ZDDs used in the experiments. For the Hamiltonian-cycle setting (HAMILTON), we used att48 (ATT) and dantzig42 (DANTZIG) datasets in TSPLIB [48]. Following [11, 44], we obtained networks in Figure 4 using Delaunay triangulation [54]. For the Steiner-tree setting (STEINER), we used Uninett 2011 (UNINETT) and TW Telecom (TW) networks of Internet Topology Zoo [31]. We selected terminal vertices as shown in Figure 4. We can see in Table 1 that ZDDs are much smaller than the strategy sets. As mentioned in Section 3.2, we can construct ZDDs in a preprocessing step, and the construction times were so short as to be negligible compared with the times taken for minimizing the social cost (see Section 4.2). Therefore, we do not take the construction times into account in what follows.

Table 1: Sizes of networks $G = (V, E)$, strategy sets $\mathcal{S}$, and ZDDs $\mathsf{Z}_{\mathcal{S}}$. ZDDs for DANTZIG, ATT, UNINETT, and TW were constructed in 172, 258, 4, and 6 ms, respectively.

| STRATEGY | DATASET | $|V|$ | $|E|$ | $|\mathcal{S}|$ | $|\mathsf{Z}_{\mathcal{S}}|$ |
|---|---|---|---|---|---|
| HAMILTON | DANTZIG | 42 | 115 | 15164782028 ($\geq 1.5 \times 10^{10}$) | 23479 |
| | ATT | 48 | 130 | 1041278451879 ($\geq 1.0 \times 10^{12}$) | 35388 |
| STEINER | UNINETT | 69 | 96 | 88920985482584429311488 ($\geq 8.8 \times 10^{22}$) | 3284 |
| | TW | 76 | 115 | 71363851011296173824385276416 ($\geq 7.1 \times 10^{28}$) | 5583 |

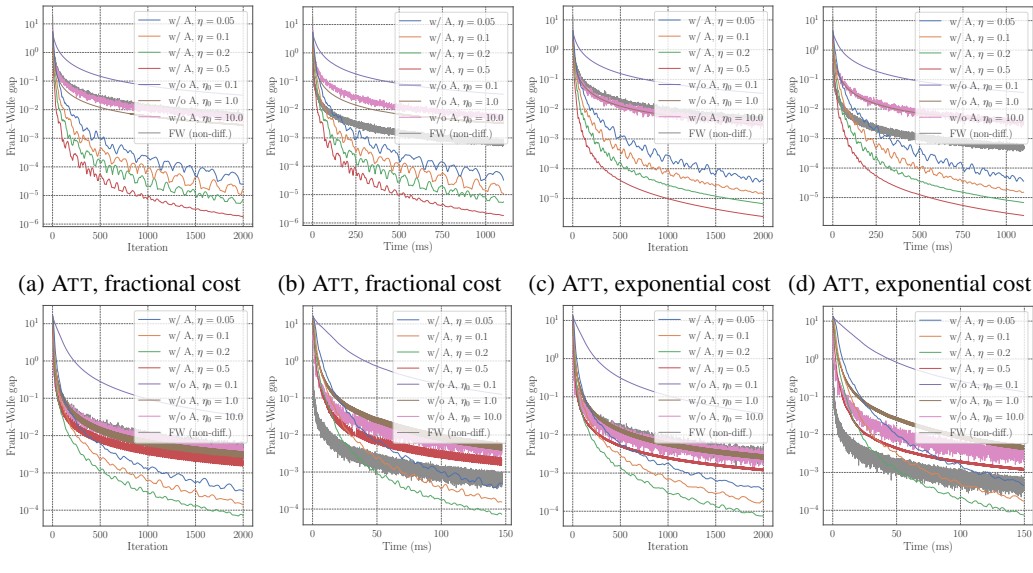

(a) ATT, fractional cost    (b) ATT, fractional cost    (c) ATT, exponential cost    (d) ATT, exponential cost

(e) TW, fractional cost    (f) TW, fractional cost    (g) TW, exponential cost    (h) TW, exponential cost

Figure 2: Convergence results of equilibrium computation methods, where w/ A, w/o A, and FW represent Algorithm 1, naive differentiable Frank–Wolfe without acceleration, and standard non-differentiable Frank–Wolfe, respectively. We present the results on the other settings in Appendix C.

## 4.1 Empirical convergence of equilibrium computation

We studied the empirical convergence of Algorithm 1 with acceleration (w/ A), where we let $\eta = 0.05$, 0.1, 0.2, and 0.5. We applied it to the minimization problems of form $\min_{\boldsymbol{y} \in \mathcal{C}} f(\boldsymbol{y}; \boldsymbol{\theta})$, where $f$ is a potential function defined by cost function $c_i(y_i; \boldsymbol{\theta})$ (fractional or exponential). We let $\boldsymbol{\theta} = \mathbf{1}$.

For comparison, we used two kinds of baselines. One is a differentiable Frank–Wolfe algorithm without acceleration (w/o A), which just replaces $\mathrm{argmin}$ with softmin as explained in Section 2.1. To guarantee the convergence of the modified algorithm, we let $\eta_t = \eta_0 \times t$ ($\eta_0 = 0.1$, 1.0, and 10.0). The other is the standard non-differentiable Frank–Wolfe algorithm (FW) implemented as in [27].

Figure 2 shows how quickly the Frank–Wolfe gap [27], which is an upper bound of the objective error, decreased as the number of iterations and the computation time increased. w/ A and w/o A tend to be faster and slower than FW, respectively. That is, Algorithm 1 (w/ A) becomes both differentiable and faster than the original FW, while the naive modified one (w/o A) becomes differentiable but slower. As in the TW results, however, w/ A with a too large $\eta$ ($\eta = 0.5$) sometimes failed to be accelerated; this is reasonable since Theorem 1 requires $\eta$ to be a moderate value. Thus, if we can locate appropriate $\eta$, Algorithm 1 achieves faster convergence in practice. As discussed in Section 2.4, we can search for $\eta$ by examining the empirical convergence for various $\eta$ values, as we did above.

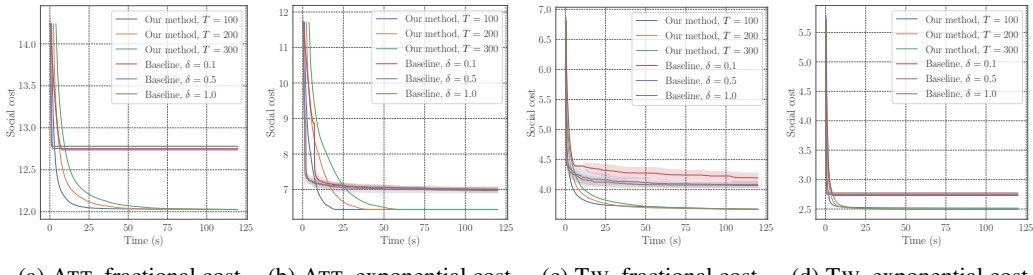

(a) ATT, fractional cost  (b) ATT, exponential cost  (c) TW, fractional cost  (d) TW, exponential cost

Figure 3: Plots of social costs achieved on network design instances. Error bands of baseline methods show standard deviations over 20 trials. We present results on other settings in Appendix C.

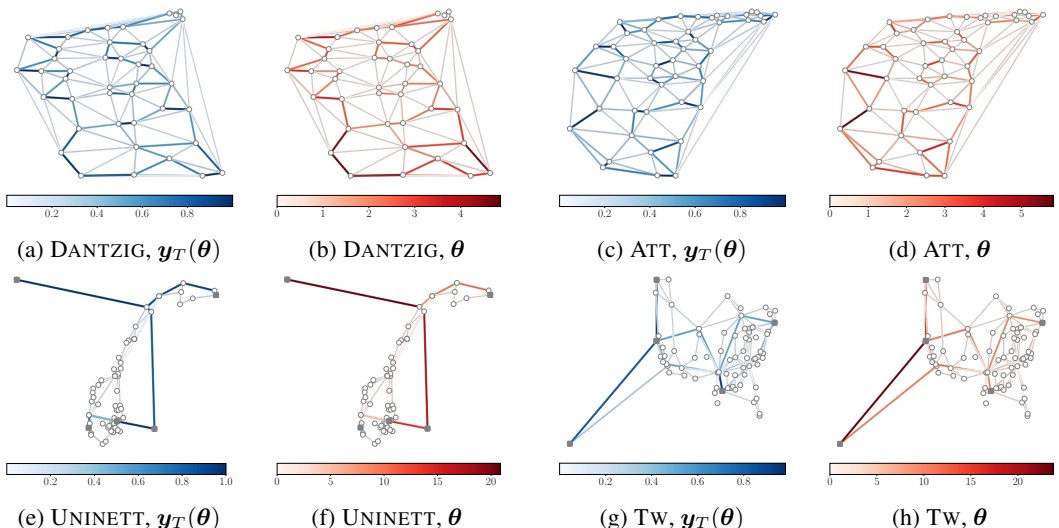

(a) DANTZIG, $\boldsymbol{y}_T(\boldsymbol{\theta})$  (b) DANTZIG, $\boldsymbol{\theta}$  (c) ATT, $\boldsymbol{y}_T(\boldsymbol{\theta})$  (d) ATT, $\boldsymbol{\theta}$

(e) UNINETT, $\boldsymbol{y}_T(\boldsymbol{\theta})$  (f) UNINETT, $\boldsymbol{\theta}$  (g) TW, $\boldsymbol{y}_T(\boldsymbol{\theta})$  (h) TW, $\boldsymbol{\theta}$

Figure 4: Network illustrations: five square vertices in UNINETT and TW indicate terminals. Blue (a, c, e, g) and red (b, d, f, h) edges represent $\boldsymbol{y}_T(\boldsymbol{\theta})$ and $\boldsymbol{\theta}$ values, respectively, computed by our method with $\eta = 0.1$ and $T = 300$ for fractional-cost instances.

## 4.2 Stackelberg models for designing communication networks

We consider minimizing social cost $F(\boldsymbol{\theta}, \boldsymbol{y}(\boldsymbol{\theta}))$. To this end, roughly speaking, we should assign large $\theta_i$ values to edges with large $y_i$ values. We applied the projected gradient method with a step size of 5.0 to problem (1). We approximated $\nabla F(\boldsymbol{\theta}, \boldsymbol{y}(\boldsymbol{\theta}))$ by applying automatic differentiation to $F(\boldsymbol{\theta}, \boldsymbol{y}_T(\boldsymbol{\theta}))$, where $\boldsymbol{y}_T(\boldsymbol{\theta})$ was computed by Algorithm 1 with $T = 100$, 200, and 300.

To the best of our knowledge, no existing methods can efficiently deal with the problems considered here due to the complicated structures of $\mathcal{S}$. Therefore, as a baseline method, we used the following iterative heuristic. Given current $\boldsymbol{y}(\boldsymbol{\theta})$, we replace $\boldsymbol{\theta}$ with $\boldsymbol{\theta} + \delta(\boldsymbol{y}(\boldsymbol{\theta}) - \bar{y}(\boldsymbol{\theta})\mathbf{1})$, where $\delta > 0$ and $\bar{y}(\boldsymbol{\theta}) = \frac{1}{n}\sum_{i\in[n]} y_i(\boldsymbol{\theta})$. That is, we increase/decrease $\theta_i$ if edge $i$ is used more/less than average. We then project $\boldsymbol{\theta}$ onto $\Theta$ and compute $\boldsymbol{y}(\boldsymbol{\theta})$ with Algorithm 1 ($T = 300$). If $F(\boldsymbol{\theta}, \boldsymbol{y}(\boldsymbol{\theta}))$ value does not decrease after the above update, we restart from a random point in $\Theta$.

Figure 3 compares our method ($\eta = 0.1$) and the baseline on ATT and TW instances, where both started from $\boldsymbol{\theta} = \mathbf{1}$ and continued to update $\boldsymbol{\theta}$ for two minutes. Our method found better $\boldsymbol{\theta}$ values than the baseline. The results only imply the empirical tendency, and our method is not guaranteed to find globally optimal $\boldsymbol{\theta}$. Nevertheless, experiments in Section 4.3 show that it tends to find a global optimum at least for small instances. Figure 4 shows the $\boldsymbol{y}_T(\boldsymbol{\theta})$ and $\boldsymbol{\theta}$ values obtained by our method with $\eta = 0.1$ and $T = 300$ for fractional-cost instance. We confirmed that large $\theta_i$ values were successfully assigned to edges with large $y_i$ values. Those results demonstrate that our method is useful for addressing Stackelberg models of CCGs with complicated combinatorial strategy sets $\mathcal{S}$.

### 4.3 Experiments on empirical convergence to global optimum

We performed additional experiments on small instances to see whether the projected gradient method used in Section 4.2 can empirically find $\boldsymbol{\theta}$ that is close to being optimal. We used small selfish-routing instances, where a graph is given by the left one in Figure 1 and the edges are numbered from 1 to 5 in that order. We let strategy set $\mathcal{S}$ be the set of all simple $s$-$t$ paths. The cost functions and feasible region $\Theta$ were set as with those in the above sections. Our goal is to minimize social cost $F(\boldsymbol{\theta}, \boldsymbol{y}(\boldsymbol{\theta}))$.

As in Section 4.2, we computed $\boldsymbol{y}(\boldsymbol{\theta})$ using Algorithm 1 with $\eta = 0.1$ and $T = 300$, and performed the projected gradient descent to minimize $F(\boldsymbol{\theta}, \boldsymbol{y}(\boldsymbol{\theta}))$, where gradient $\nabla F(\boldsymbol{\theta}, \boldsymbol{y}(\boldsymbol{\theta}))$ was computed with backpropagation. On the other hand, to obtain (approximations of) globally optimal $\boldsymbol{\theta}$, we performed an exhaustive search over the feasible region, where the step size was set to $0.05$. Regarding computation times, the projected gradient method converged in less than 30 iterations, which took less than 20 ms, while the exhaustive search took about 1000 seconds.

**Results on fractional costs.** A global optimum found by the exhaustive search was $\boldsymbol{\theta} = (0, 2.5, 0, 0, 2.5)$, whose social cost $F(\boldsymbol{\theta}, \boldsymbol{y}(\boldsymbol{\theta}))$ was $6.444$. Our method started from $\boldsymbol{\theta} = \mathbf{1}$, whose social cost was $7.000$, and returned $\boldsymbol{\theta} = (1.25, 1.25, 0, 1.25, 1.25)$ with social cost $6.444$. Although the solution is different from that of the exhaustive search, both attain the identical social cost. Thus, the solution returned by the projected gradient method is also globally optimal.

**Results on exponential costs.** A global optimum found by the exhaustive search was $\boldsymbol{\theta} = (0, 2.5, 0, 0, 2.5)$ with social cost $3.517$. Our method started from $\boldsymbol{\theta} = \mathbf{1}$ with social cost $5.678$ and reached $\boldsymbol{\theta} = (0, 2.5, 0, 0, 2.5)$ with social cost $3.517$. Along the way, the projected gradient method was about to be trapped in $\boldsymbol{\theta} = (1.25, 1.25, 0, 1.25, 1.25)$ with social cost $4.865$, which seems to be a saddle point. However, it successfully got out of there and reached the global optimum.

## 5 Conclusion and discussion

We proposed a differentiable pipeline that connects CCG parameters to their equilibria, enabling us to apply gradient-based methods to the Stackelberg models of CCGs. Our Algorithm 1 leverages softmin to make the Frank–Wolfe algorithm both differentiable and faster. ZDD-based softmin computation (Algorithm 2) enables us to efficiently deal with complicated CCGs. It also naturally works with automatic differentiation, offering an easy way to compute desired derivatives. Experiments confirmed the accelerated empirical convergence and practicality of our method.

An important future direction is further studying theoretical aspects. From our experimental results, $\nabla \boldsymbol{y}_T(\boldsymbol{\theta})$ is expected to converge to $\nabla \boldsymbol{y}(\boldsymbol{\theta})$, although its theoretical analysis is very difficult. Recently, some relevant results have been obtained for simple cases where iterative optimization methods are written by a contraction map defined on an unconstrained domain [1, 20]. In our CCG cases, however, we need to study iterative algorithms that numerically solve the constrained potential minimization, which requires a more profound understanding of iterative differentiation approaches. Another interesting future work is to make linearly convergent Frank–Wolfe variants [33] differentiable.

Finally, we discuss limitations and possible negative impacts. Our work does not cover cases where minimizer $\boldsymbol{y}(\boldsymbol{\theta})$ of potential functions is not unique. Since the complexity of our method mainly depends on the ZDD sizes, it does not work if ZDDs are prohibitively large, which can happen when strategy sets consist of the substructures of dense networks. Nevertheless, many real-world networks are sparse, and thus our ZDD-based method is often effective, as demonstrated in experiments. At a meta-level, optimizing social infrastructures in terms of a single objective function (e.g., the social cost) may lead to an extreme choice that is detrimental to some individuals. We hope our method can provide a basis for designing social infrastructures that are beneficial for all.

## Acknowledgements

The authors thank the anonymous reviewers for their valuable feedback, corrections, and suggestions. This work was partially supported by JST ERATO Grant Number JPMJER1903 and JSPS KAKENHI Grant Number JP20H05963.

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
