# Appendix

## A    Extension to asymmetric CCGs

First, we introduce additional notation and definitions for extending our problem setting to the case with asymmetric CCGs, where there are multiple types of strategy sets. We can recover the simpler symmetric setting, which we studied in the main paper, by modifying the notation presented below: $r = 1$, $\mathcal{S}^1 = \mathcal{S}$, and $m^1 = 1$.

### A.1    Problem setting

There are $r$ populations of players, whose strategy sets are $\mathcal{S}^1, \ldots, \mathcal{S}^r \subseteq 2^{[n]}$. Let $d^p := |\mathcal{S}^p|$ for $p \in [r]$ and define $d := d^1 + \cdots + d^r$. Let $\boldsymbol{z}^p \in \triangle^{d^p}$ for each $p \in [r]$ and $\boldsymbol{z} = (\boldsymbol{z}^1, \ldots, \boldsymbol{z}^r) \in \mathcal{D} := \triangle^{d^1} \times \cdots \times \triangle^{d^r}$. Let $z_S^p \in [0, 1]$ be the entry of $\boldsymbol{z}^p$ corresponding to $S \in \mathcal{S}^p$, which indicates the proportion of players in the $p$-th group who choose strategy $S \in \mathcal{S}^p$.

For each $p \in [r]$, let $\boldsymbol{\Lambda}^p \in \{0, 1\}^{n \times d^p}$ be a matrix whose $(i, S)$ entry is 1 iff $i \in [n]$ is included in $S \in \mathcal{S}^p$; the columns of $\boldsymbol{\Lambda}^p$ consist of $\{\boldsymbol{1}_S\}_{S \in \mathcal{S}^p}$. Let $\boldsymbol{x}^p = \sum_{S \in \mathcal{S}^p} z_S^p \boldsymbol{1}_S = \boldsymbol{\Lambda}^p \boldsymbol{z}^p \in [0, 1]^n$, whose $i$-th entry is the proportion of players in $p$ who choose $S \in \mathcal{S}^p$ such that $i \in S$. Note that $\boldsymbol{x}^p$ is in the convex hull, $\mathrm{conv}(\mathcal{S}^p) := \{ \sum_{S \in \mathcal{S}} z_S^p \boldsymbol{1}_S \mid \boldsymbol{z}^p \in \triangle^{d^p} \}$.

For each $p \in [r]$, let $m^p > 0$ be the total mass of players in the $p$-th group. We define $\boldsymbol{\Lambda} := [m^1 \boldsymbol{\Lambda}^1, \ldots, m^r \boldsymbol{\Lambda}^r] \in \mathbb{R}^{n \times d}$ and let $\boldsymbol{y}$ be a vector whose $i$-th entry indicates the total mass of players using $i \in [n]$, i.e., $\boldsymbol{y} = \sum_{p \in [r]} m^p \boldsymbol{x}^p = \sum_{p \in [r]} m^p \boldsymbol{\Lambda}^p \boldsymbol{z}^p = \boldsymbol{\Lambda} \boldsymbol{z}$. Note that for any $\boldsymbol{z} \in \mathcal{D}$, $\boldsymbol{y} = \boldsymbol{\Lambda} \boldsymbol{z}$ is always included in $\mathcal{C} := \{ \sum_{p \in [r]} m^p \boldsymbol{x}^p \mid \boldsymbol{x}^p \in \mathrm{conv}(\mathcal{S}^p) \text{ for } p \in [r] \}$.

Analogous to the symmetric case, each $i \in [n]$ has cost function $c_i(\cdot; \boldsymbol{\theta})$, where we assume $c_i(y_i; \boldsymbol{\theta})$ to be strictly increasing in $y_i$ for any $\boldsymbol{\theta} \in \Theta$ and differentiable in $\boldsymbol{\theta}$ for any $\boldsymbol{y} \in \mathcal{C}$. A player choosing strategy $S$ incurs cost $c_S(\boldsymbol{y}; \boldsymbol{\theta}) := \sum_{i \in S} c_i(y_i; \boldsymbol{\theta})$. In the asymmetric setting, once $\boldsymbol{\theta}$ is fixed, an equilibrium is defined as follows: $\boldsymbol{z} \in \mathcal{D}$ attains an equilibrium if for every $p \in [r]$, every $S \in \mathcal{S}^p$ such that $z_S^p > 0$ satisfies $c_S(\boldsymbol{y}; \boldsymbol{\theta}) \le \min_{S' \in \mathcal{S}^p} c_{S'}(\boldsymbol{y}; \boldsymbol{\theta})$ for $\boldsymbol{y} = \boldsymbol{\Lambda} \boldsymbol{z}$. That is, for every $p \in [r]$, no player in the $p$-th group is motivated to change his/her strategy in $\mathcal{S}^p$.

As with the symmetric case, $\boldsymbol{z} \in \mathcal{D}$ attains an equilibrium iff $\boldsymbol{y} = \boldsymbol{\Lambda} \boldsymbol{z}$ is a (unique) minimizer of the following potential function minimization:

$$\underset{\boldsymbol{y}}{\text{minimize }} f(\boldsymbol{y}; \boldsymbol{\theta}) \quad \text{subject to } \boldsymbol{y} \in \mathcal{C},$$

where $f(\boldsymbol{y}; \boldsymbol{\theta}) = \sum_{i \in [n]} \int_0^{y_i} c_i(y; \boldsymbol{\theta}) \mathrm{d}y$ is a parameterized potential function. In what follows, we also consider the following formulation of the above problem:

$$\underset{\boldsymbol{z}}{\text{minimize }} \Phi(\boldsymbol{z}; \boldsymbol{\theta}) := f(\boldsymbol{\Lambda} \boldsymbol{z}; \boldsymbol{\theta}) \quad \text{subject to } \boldsymbol{z} \in \mathcal{D}. \tag{A1}$$

Since $d$ is exponential in $n$ in general, we cannot directly deal with problem (A1) in practice. We only use the formulation for the theoretical analysis; our method does not explicitly deal with (A1).

### A.2    Extension of our algorithm

To address the asymmetric setting, we need to slightly modify our algorithm. Algorithm A1 presents details of the modified algorithm. In Step 5, we use softmin oracle $\boldsymbol{\mu}_{\mathcal{S}^p}$ for each $p \in [r]$ to obtain $\boldsymbol{x}^p$, and in Step 6 we aggregate them to obtain $\boldsymbol{x}_t = \sum_{p \in [r]} m^p \boldsymbol{x}_t^p$. In the symmetric setting, where $r = 1$, $\mathcal{S}^1 = \mathcal{S}$, and $m^1 = 1$, Algorithm A1 is equivalent to Algorithm 1. For each $p \in [r]$, softmin oracle $\boldsymbol{\mu}_{\mathcal{S}^p}$ returns the following $n$-dimensional vector:

$$\boldsymbol{\mu}_{\mathcal{S}^p}(\boldsymbol{c}) = \sum_{S \in \mathcal{S}^p} \boldsymbol{1}_S \frac{\exp\left(-\boldsymbol{c}^\top \boldsymbol{1}_S\right)}{\sum_{S' \in \mathcal{S}^p} \exp\left(-\boldsymbol{c}^\top \boldsymbol{1}_{S'}\right)},$$

where $\boldsymbol{c} \in \mathbb{R}^n$. To compute softmin for each $p \in [r]$ with ZDDs, we construct $\mathsf{Z}_{\mathcal{S}^1}, \ldots, \mathsf{Z}_{\mathcal{S}^r}$ in a preprocessing step and use Algorithm 2. As with the symmetric setting, once $\mathsf{Z}_{\mathcal{S}^p}$ is constructed for each $p \in [r]$, we can repeatedly use it every time $\boldsymbol{\mu}_{\mathcal{S}^p}$ is called.

**Algorithm A1** Differentiable Frank–Wolfe-based equilibrium computation for asymmetric settings

1: $c_0 = 0$, $s_0 = 0$, $x_{-1} = x_0 = \sum_{p \in [r]} m^p \mu_{\mathcal{S}^p}(m^p c_0)$, and $\alpha_t = t$ $(t = 0, \ldots, T)$
2: **for** $t = 1, \ldots, T$ :
3:      $s_t = s_{t-1} - \alpha_{t-1} x_{t-2} + (\alpha_{t-1} + \alpha_t) x_{t-1}$
4:      $c_t = c_{t-1} + \eta \alpha_t \nabla f \left( \frac{2}{t(t+1)} s_t; \boldsymbol{\theta} \right)$
5:      $x_t^p = \mu_{\mathcal{S}^p}(m^p c_t)$ for each $p \in [r]$
6:      $x_t = \sum_{p \in [r]} m^p x_t^p$
    **return** $y_T(\boldsymbol{\theta}) = \frac{2}{T(T+1)} \sum_{t=1}^{T} \alpha_t x_t$

# B Proof of Theorem 1

We prove the following convergence guarantee of Algorithm A1.

**Theorem A1.** *If $\eta \in [\frac{1}{CL}, \frac{1}{4L}]$ holds for some constant $C > 4$, we have*

$$f(\boldsymbol{y}_T(\boldsymbol{\theta}); \boldsymbol{\theta}) - \min_{\boldsymbol{y} \in \mathcal{C}} f(\boldsymbol{y}; \boldsymbol{\theta}) \leq \mathrm{O} \left( \frac{CL \sum_{p \in [r]} \ln d^p}{T^2} \right).$$

Here $L$ is the smoothness parameter of $\Phi$ defined in (A1). I.e., for $\Phi(\boldsymbol{z}; \boldsymbol{\theta}) := f(\boldsymbol{\Lambda} \boldsymbol{z}; \boldsymbol{\theta})$, we assume that $\Phi(\boldsymbol{z}'; \boldsymbol{\theta}) \leq \Phi(\boldsymbol{z}; \boldsymbol{\theta}) + \langle \nabla \Phi(\boldsymbol{z}; \boldsymbol{\theta}), \boldsymbol{z}' - \boldsymbol{z} \rangle + \frac{L}{2} \|\boldsymbol{z}' - \boldsymbol{z}\|^2$ holds for all $\boldsymbol{z}, \boldsymbol{z}' \in \mathbb{R}^d$. Note that by setting $r = 1$ as explained in Appendix A, we can recover the converge guarantee of the symmetric setting (Theorem 1 in Section 2.3). Below we fix $\boldsymbol{\theta} \in \Theta$ and omit $\boldsymbol{\theta}$ for simplicity.

The following analysis is based on [59]. Our technical contribution is to reveal that their accelerated algorithm (Algorithm A2) can be used as a differentiable Frank–Wolfe algorithm (Algorithm A1), which explicitly accepts backpropagation and enjoys efficient ZDD-based implementation. Note that since Wang and Abernethy [59] did not mention the differentiability of Algorithm A2, our work is the first to show that the Frank–Wolfe algorithm can simultaneously be made differentiable and faster.

We introduce some notation and definitions. We define the Kullback–Leibler (KL) divergence as $D_{\mathrm{KL}}(\boldsymbol{z}; \boldsymbol{z}') := \langle \boldsymbol{z}, \ln(\boldsymbol{z}/\boldsymbol{z}') \rangle$ $(\forall \boldsymbol{z}, \boldsymbol{z}' \in \mathcal{D})$, where $\boldsymbol{z}/\boldsymbol{z}'$ and $\ln$ are an element-wise division and a logarithm, respectively. Let $\alpha_t = t$ and denote the sequence of $\alpha_t$s by $\boldsymbol{\alpha}_{1:t} = \alpha_1, \ldots, \alpha_{t-1}, \alpha_t$. We also define modified sequence $\boldsymbol{\alpha}'_{1:t-1} = \alpha'_1, \ldots, \alpha'_{t-1}$ so that $\alpha'_s = \alpha_s$ $(s \leq t-2)$ and $\alpha'_{t-1} = \alpha_{t-1} + \alpha_t$ hold, where $\boldsymbol{\alpha}'_{1:0} = \alpha_0 = 0$. Let $A_t = \sum_{s=1}^{t} \alpha_s$. For any sequence of vectors $\boldsymbol{u}_0, \ldots, \boldsymbol{u}_t$, we define $\boldsymbol{u}_{\boldsymbol{\alpha}_{1:t}} = \sum_{s=1}^{t} \alpha_s \boldsymbol{u}_s$, $\bar{\boldsymbol{u}}_{\boldsymbol{\alpha}_{1:t}} = \frac{1}{A_t} \sum_{s=1}^{t} \alpha_s \boldsymbol{u}_s$, $\boldsymbol{u}_{\boldsymbol{\alpha}'_{1:t-1}} = \sum_{s=1}^{t-1} \alpha'_s \boldsymbol{u}_s$, and $\bar{\boldsymbol{u}}_{\boldsymbol{\alpha}'_{1:t-1}} = \frac{1}{A_t} \sum_{s=1}^{t-1} \alpha'_s \boldsymbol{u}_s$, where we let $\boldsymbol{u}_{\boldsymbol{\alpha}'_{1:0}} = \bar{\boldsymbol{u}}_{\boldsymbol{\alpha}'_{1:0}} = \boldsymbol{u}_0$.

To prove Theorem A1, we use the following relationship between Algorithms A1 and A2.

**Lemma A1.** *For $\boldsymbol{x}_0, \ldots, \boldsymbol{x}_T$ and $\boldsymbol{z}_0, \ldots, \boldsymbol{z}_T$ obtained in Step 6 of Algorithm A1 and Step 4 of Algorithm A2, respectively, we have $\boldsymbol{x}_t = \boldsymbol{\Lambda} \boldsymbol{z}_t$ $(t = 0, \ldots, T)$.*

*Proof of Lemma A1.* We first show that for each $p \in [r]$, $\boldsymbol{z}_t^p \in \triangle^{d^p}$ obtained by Algorithm A2 satisfies

$$\boldsymbol{z}_t^p \propto \exp \left( -\sum_{s=1}^{t} \eta \alpha_s \boldsymbol{g}_s^p \right) \qquad (t = 1, \ldots, T),$$

where $\exp$ is taken in an element-wise manner. From the KKT condition of $\operatorname{argmin}$ in Step 4 of Algorithm A2, for each $p \in [r]$, we have

$$\alpha_t \boldsymbol{g}_t^p + \frac{1}{\eta}(\ln \boldsymbol{z}^p + \mathbf{1} - \ln \boldsymbol{z}_{t-1}^p) - \nu^p \mathbf{1} = 0,$$

where $\nu^p \in \mathbb{R}$ is a multiplier corresponding to the equality constraint, $\mathbf{1}^\top \boldsymbol{z}_t^p = 1$. Note that we need not take inequality constraint $\boldsymbol{z}^p \geq 0$ into account since entropic regularization forces $\boldsymbol{z}_S^p$ to be positive. The above equality implies that entries in $\boldsymbol{z}_t^p$ are proportional to those of $\exp(-\eta \alpha_t \boldsymbol{g}_t^p + \ln \boldsymbol{z}_{t-1}^p)$, and thus we obtain $\boldsymbol{z}_t^p \propto \boldsymbol{z}_0^p \odot \exp(-\sum_{s=1}^{t} \eta \alpha_s \boldsymbol{g}_s^p)$ by induction, where $\odot$ is the element-wise product. Since $\boldsymbol{z}_0^p = (1/d^p, \ldots, 1/d^p)$, we get $\boldsymbol{z}_t^p \propto \exp(-\sum_{s=1}^{t} \eta \alpha_s \boldsymbol{g}_s^p)$.

---

**Algorithm A2** Accelerated Frank–Wolfe algorithm as a two-player game [59]

1: $\boldsymbol{z}_0 = (\boldsymbol{z}_0^1, \ldots, \boldsymbol{z}_0^r)$ where $\boldsymbol{z}_0^p = (1/d^p, \ldots, 1/d^p) \in \triangle^{d^p}$ for each $p \in [r]$
2: **for** $t = 1, \ldots, T$ :
3:      $\boldsymbol{g}$-player's action: $\boldsymbol{g}_t = \nabla\Phi\left(\bar{\boldsymbol{z}}_{\boldsymbol{\alpha'}_{1:t-1}}\right)$     $\triangleright$ I.e., $\boldsymbol{g}_t = \mathrm{argmin}_{\boldsymbol{g}\in\mathbb{R}^d} \Phi^*(\boldsymbol{g}) - \langle\bar{\boldsymbol{z}}_{\boldsymbol{\alpha'}_{1:t-1}}, \boldsymbol{g}\rangle$
4:      $\boldsymbol{z}$-player's action: $\boldsymbol{z}_t = \mathrm{argmin}_{\boldsymbol{z}\in\mathcal{D}}\langle\alpha_t\boldsymbol{g}_t, \boldsymbol{z}\rangle + \frac{1}{\eta}D_{\mathrm{KL}}(\boldsymbol{z}; \boldsymbol{z}_{t-1})$
    **return** $\bar{\boldsymbol{z}}_{\boldsymbol{\alpha}_{1:T}}$

---

We then show by induction that Algorithm A1 computes $\boldsymbol{x}_t$ that satisfies $\boldsymbol{x}_t = \boldsymbol{\Lambda}\boldsymbol{z}_t$, where $\boldsymbol{z}_t^p \propto \exp(-\sum_{s=1}^t \eta\alpha_s\boldsymbol{g}_s^p)$ holds for $t \geq 1$ as shown above. The base case of $t = 0$ can be confirmed as follows. Since $\boldsymbol{c}_0 = 0$, we have

$$\boldsymbol{\mu}_{\mathcal{S}^p}(m^p\boldsymbol{c}_0) = \sum_{S\in\mathcal{S}^p} \mathbf{1}_S \frac{\exp\left(-m^p\boldsymbol{c}_0^\top\mathbf{1}_S\right)}{\sum_{S'\in\mathcal{S}^p}\exp\left(-m^p\boldsymbol{c}_0^\top\mathbf{1}_{S'}\right)} = \sum_{S\in\mathcal{S}^p}\mathbf{1}_S\frac{1}{d^p} = \boldsymbol{\Lambda}^p\boldsymbol{z}_0^p,$$

which implies

$$\boldsymbol{x}_0 = \sum_{p\in[r]} m^p\boldsymbol{\mu}_{\mathcal{S}^p}(m^p\boldsymbol{c}_0) = \sum_{p\in[r]} m^p\boldsymbol{\Lambda}^p\boldsymbol{z}_0^p = \boldsymbol{\Lambda}\boldsymbol{z}_0.$$

We then assume that $\boldsymbol{x}_s = \boldsymbol{\Lambda}\boldsymbol{z}_s$ holds for $s = 0, \ldots, t-1$. In the $t$-th step, Algorithm A1 computes

$$\boldsymbol{s}_t = \boldsymbol{x}_{\boldsymbol{\alpha'}_{1:t-1}} \qquad \text{and} \qquad \boldsymbol{c}_t = \boldsymbol{c}_0 + \sum_{s=1}^t \eta\alpha_s\nabla f(\bar{\boldsymbol{x}}_{\boldsymbol{\alpha'}_{1:s-1}}) = \sum_{s=1}^t \eta\alpha_s\nabla f(\bar{\boldsymbol{x}}_{\boldsymbol{\alpha'}_{1:s-1}}).$$

From the induction hypothesis, it holds that $\frac{2}{t(t+1)}\boldsymbol{s}_t = \bar{\boldsymbol{x}}_{\boldsymbol{\alpha'}_{1:t-1}} = \boldsymbol{\Lambda}\bar{\boldsymbol{z}}_{\boldsymbol{\alpha'}_{1:t-1}}$, which implies

$$\nabla\Phi(\bar{\boldsymbol{z}}_{\boldsymbol{\alpha'}_{1:s-1}}) = \boldsymbol{\Lambda}^\top\nabla f(\bar{\boldsymbol{x}}_{\boldsymbol{\alpha'}_{1:s-1}}).$$

With these equations, we obtain

$$\boldsymbol{\Lambda}^\top\boldsymbol{c}_t = \sum_{s=1}^t \eta\alpha_s\boldsymbol{\Lambda}^\top\nabla f(\bar{\boldsymbol{x}}_{\boldsymbol{\alpha'}_{1:s-1}}) = \sum_{s=1}^t \eta\alpha_s\nabla\Phi(\bar{\boldsymbol{z}}_{\boldsymbol{\alpha'}_{1:s-1}}) = \sum_{s=1}^t \eta\alpha_s\boldsymbol{g}_s,$$

and thus $m^p\boldsymbol{\Lambda}^{p\top}\boldsymbol{c}_t = \sum_{s=1}^t \eta\alpha_s\boldsymbol{g}_s^p$ holds for each $p \in [r]$. Therefore, for each $p \in [r]$, $\boldsymbol{x}_t^p$ computed in Algorithm A1 satisfies

$$\boldsymbol{x}_t^p = \boldsymbol{\mu}(m^p\boldsymbol{\Lambda}^{p\top}\boldsymbol{c}_t) = \boldsymbol{\Lambda}^p\frac{\exp\left(-m^p\boldsymbol{\Lambda}^{p\top}\boldsymbol{c}_t\right)}{Z_t^p} = \boldsymbol{\Lambda}^p\frac{\exp\left(-\sum_{s=1}^t \eta\alpha_s\boldsymbol{g}_s^p\right)}{Z_t^p} = \boldsymbol{\Lambda}^p\boldsymbol{z}_t^p,$$

where $Z_t^p$ is a normalizing constant and the last equality uses $\boldsymbol{z}_t^p \propto \exp(-\sum_{s=1}^t \eta\alpha_s\boldsymbol{g}_s^p)$. Hence we obtain $\boldsymbol{x}_t = \sum_{p\in[r]} m^p\boldsymbol{x}_t^p = \sum_{p\in[r]} m^p\boldsymbol{\Lambda}^p\boldsymbol{z}_t^p = \boldsymbol{\Lambda}\boldsymbol{z}_t$. Consequently, the lemma holds by induction. $\square$

Owing to Lemma A1, we can analyze the convergence of Algorithm A1 through Algorithm A2. We regard Algorithm A2 as the dynamics of a two-player zero-sum game, where $\boldsymbol{g}$-player computes $\boldsymbol{g}_t$ and $\boldsymbol{z}$-player computes $\boldsymbol{z}_t$. The payoff function of the game is a convex-linear function defined as $u(\boldsymbol{g}, \boldsymbol{z}) := \Phi^*(\boldsymbol{g}) - \langle\boldsymbol{g}, \boldsymbol{z}\rangle$, where $\Phi^*(\boldsymbol{g}) := \sup_{\boldsymbol{z}\in\mathbb{R}^d}\{\langle\boldsymbol{g}, \boldsymbol{z}\rangle - \Phi(\boldsymbol{z})\}$ is the Fenchel conjugate of $\Phi$. As detailed in [59], the players' actions are given by online optimization algorithms (in particular, $\boldsymbol{g}$-player uses a so-called *optimistic* online algorithm), and the *regret* analysis of the online algorithms yields an accelerated convergence guarantee. Formally, the following lemma holds.

**Lemma A2** ([59]). *Algorithm A2 returns $\bar{\boldsymbol{z}}_{\boldsymbol{\alpha}_{1:T}}$ satisfying $\Phi(\bar{\boldsymbol{z}}_{\boldsymbol{\alpha}_{1:T}}) - \min_{\boldsymbol{z}\in\mathcal{D}}\Phi(\boldsymbol{z}) \leq \frac{2CLB}{T(T+1)}$, where $B = D_{\mathrm{KL}}(\boldsymbol{z}^*; \boldsymbol{z}_0)$ and $\boldsymbol{z}^* \in \mathrm{argmin}_{\boldsymbol{z}\in\mathcal{D}}\Phi(\boldsymbol{z})$.*

The proof of Lemma A2 is presented in [59, Theorem 2 and Corollary 1], where $\boldsymbol{z}$-player's action is described using Bregman divergence instead of KL divergence. Since KL divergence is a special case of Bregman divergence defined with convex function $\psi(\boldsymbol{u}) = \langle\boldsymbol{u}, \ln\boldsymbol{u}\rangle$ ($\boldsymbol{u} \in \mathcal{D}$), which is 1-strongly

convex over $\mathcal{D}$, we can directly apply their result to our setting. Moreover, in their analysis, the step size is given by a non-increasing sequence, $\eta_1, \ldots, \eta_T$, to obtain a more general result. We here use a simplified version such that $\eta_1 = \cdots = \eta_T = \eta$. In this case, $B = D_{\mathrm{KL}}(\boldsymbol{z}^*; \boldsymbol{z}_0)$ appears as a leading factor as in Lemma A2 (see [59, Lemma 4] for details).

By using Lemmas A1 and A2, we can obtain Theorem A1 as follows.

*Proof of Theorem A1.* Note that we have $\boldsymbol{y}_T(\boldsymbol{\theta}) = \frac{2}{T(T+1)} \sum_{t=1}^{T} \alpha_t \boldsymbol{x}_t = \bar{\boldsymbol{y}}_{\boldsymbol{\alpha}_{1:T}}$. From Lemma A1, we have $\boldsymbol{x}_t = \boldsymbol{\Lambda} \boldsymbol{z}_t$, where $\boldsymbol{x}_t$ and $\boldsymbol{z}_t$ are those computed in Algorithms A1 and A2, respectively. Thus, we have $\boldsymbol{y}_T(\boldsymbol{\theta}) = \bar{\boldsymbol{y}}_{\boldsymbol{\alpha}_{1:T}} = \boldsymbol{\Lambda} \bar{\boldsymbol{z}}_{\boldsymbol{\alpha}_{1:T}}$. The convergence of Algorithm A2 is guaranteed by Lemma A2. Moreover, we have $B = D_{\mathrm{KL}}(\boldsymbol{z}^*; \boldsymbol{z}_0) \leq \sum_{p \in [r]} \ln d^p$ since $\boldsymbol{z}_0^p = (1/d^p, \ldots, 1/d^p) \in \triangle^{d^p}$ holds for each $p \in [r]$. Therefore, from $f(\boldsymbol{y}) = \Phi(\boldsymbol{z})$ for $\boldsymbol{y} = \boldsymbol{\Lambda} \boldsymbol{z}$, we obtain Theorem A1. $\qquad \square$

# C Full version of experiments

We present full versions of the experimental results.

## C.1 Empirical convergence of equilibrium computation

We compared the empirical convergence of three algorithms: our algorithm with acceleration (w/ A), the differentiable Frank–Wolfe algorithm without acceleration (w/o A), and the standard Frank–Wolfe algorithm (FW). We used potential minimization problems, $\min_{\boldsymbol{y} \in \mathcal{C}} f(\boldsymbol{y}; \boldsymbol{\theta})$, detailed in Section 4.1.

Figure 5 shows the results. Similar to those in Section 4.1, the performance of the differentiable Frank–Wolfe algorithm with acceleration (w/ A) tends to exceed the others (w/o A and FW), although w/ A with $\eta = 0.5$ sometimes fails to be accelerated due to the too large $\eta$ value.

## C.2 Stackelberg model for designing communication networks

We present full versions of the results on the Stackelberg model experiments described in Section 4.2.

In Figures 6 and 7, we present the social-cost results. Our method outperformed the baseline in every setting. Figures 8 and 9 present full versions of the $(\boldsymbol{y}_T(\boldsymbol{\theta}), \boldsymbol{\theta})$ illustrations. We can see that our method successfully assigned large $\theta_i$ values to the edges with large $y_i(\boldsymbol{\theta})$ values.

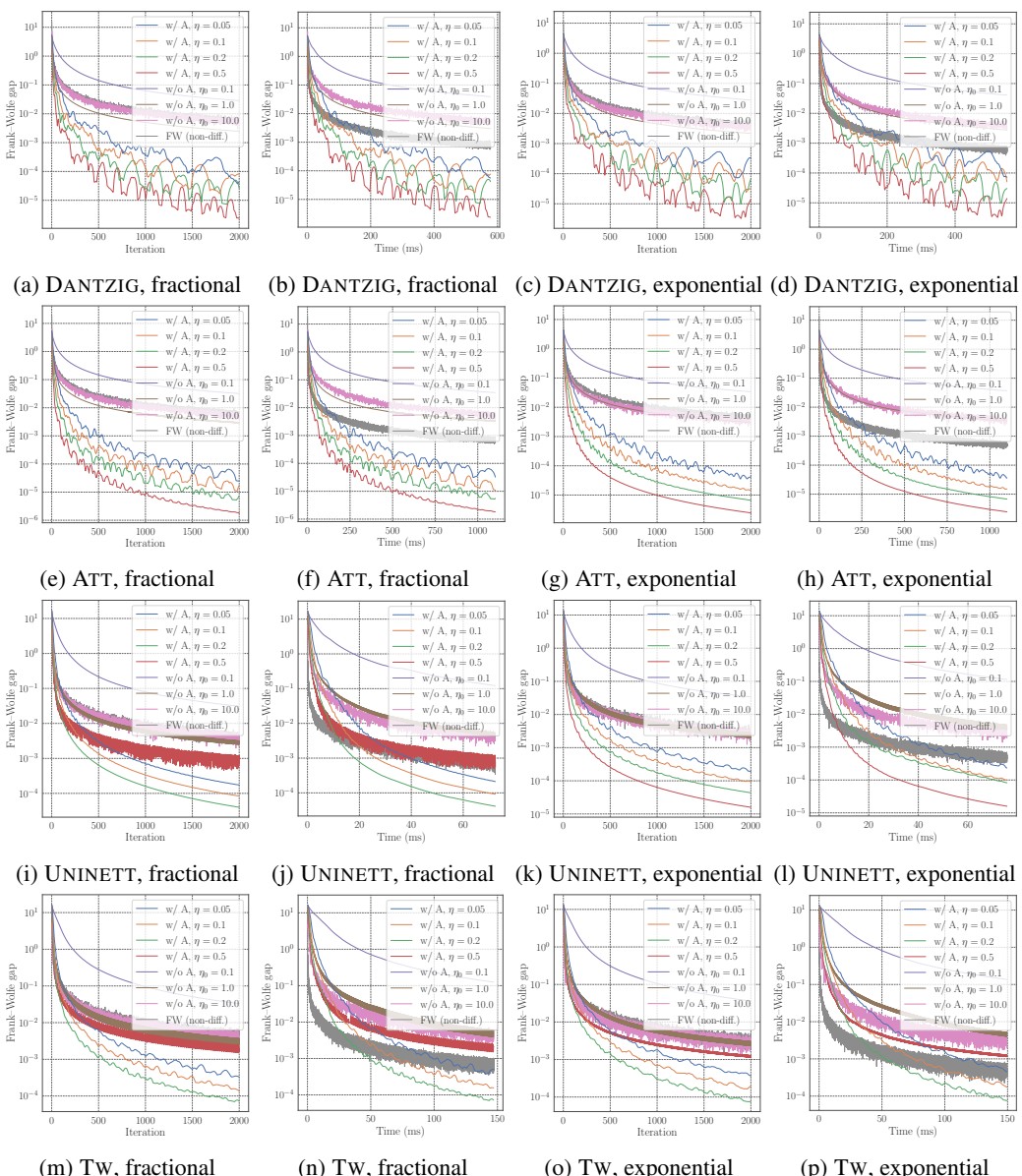

Figure 5: Convergence results on DANTZIG, ATT, UNINETT, and TW instances with fractional and exponential costs.

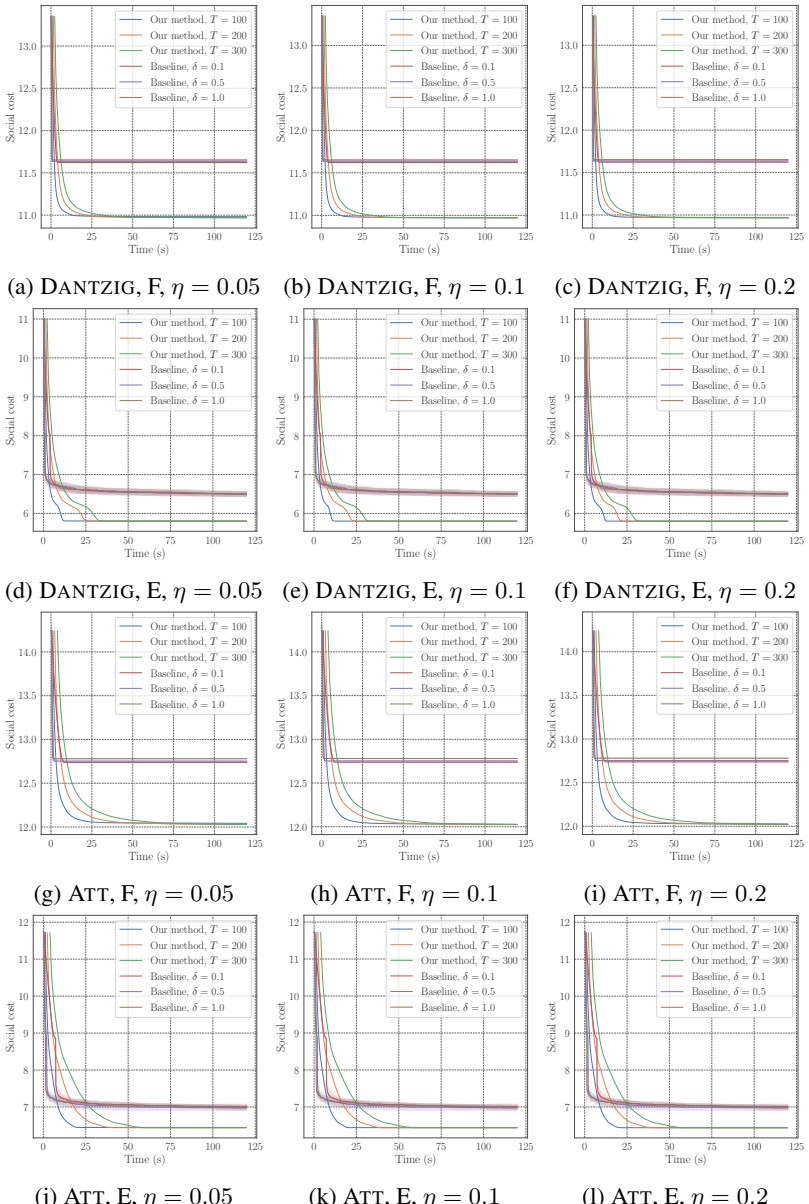

Figure 6: Plots of social costs for DANTZIG and ATT instances with fractional (F) and exponential (E) costs. We set $\eta$ of Algorithm 1 to 0.05, 0.1, and 0.2. Baseline method results are shown with means and standard deviations over 20 random trials.

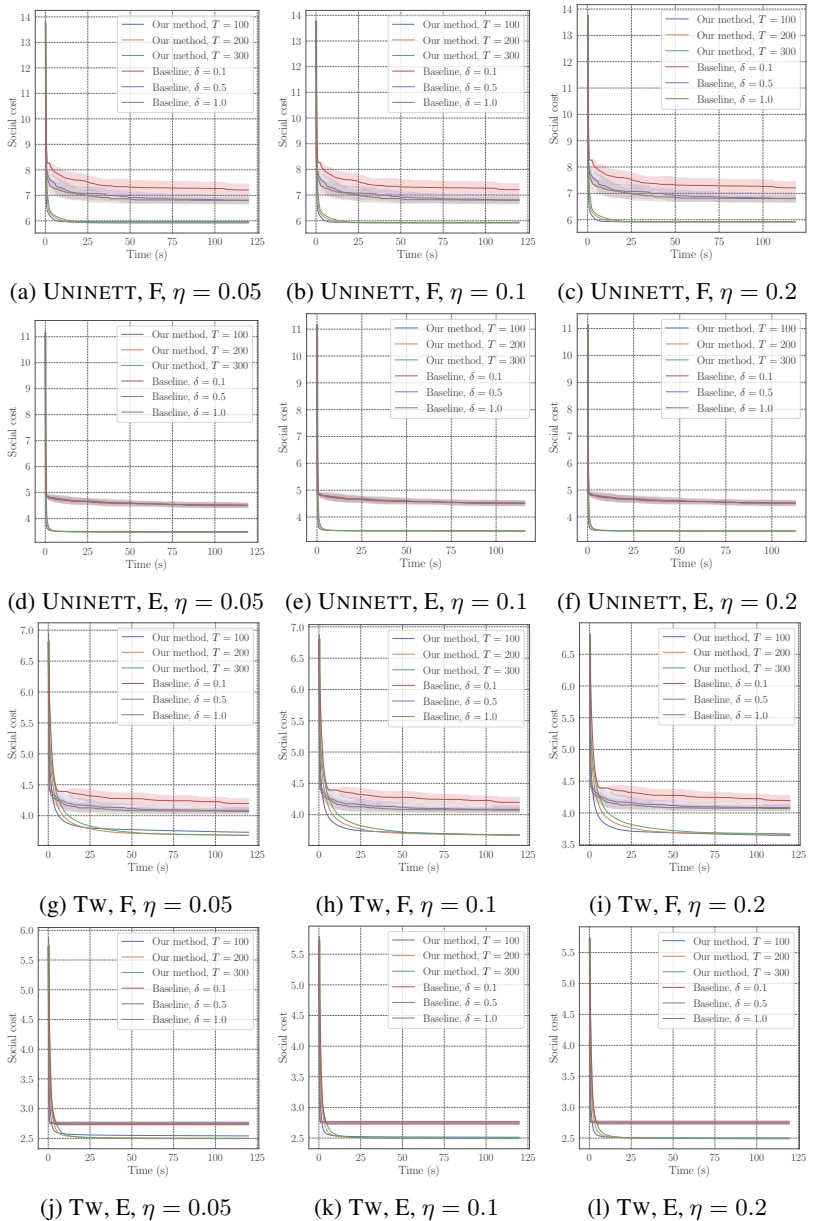

Figure 7: Plots of social costs for UNINETT and TW instances with fractional (F) and exponential (E) costs. We set $\eta$ of Algorithm 1 to 0.05, 0.1, and 0.2. Baseline method results are shown with means and standard deviations over 20 random trials.

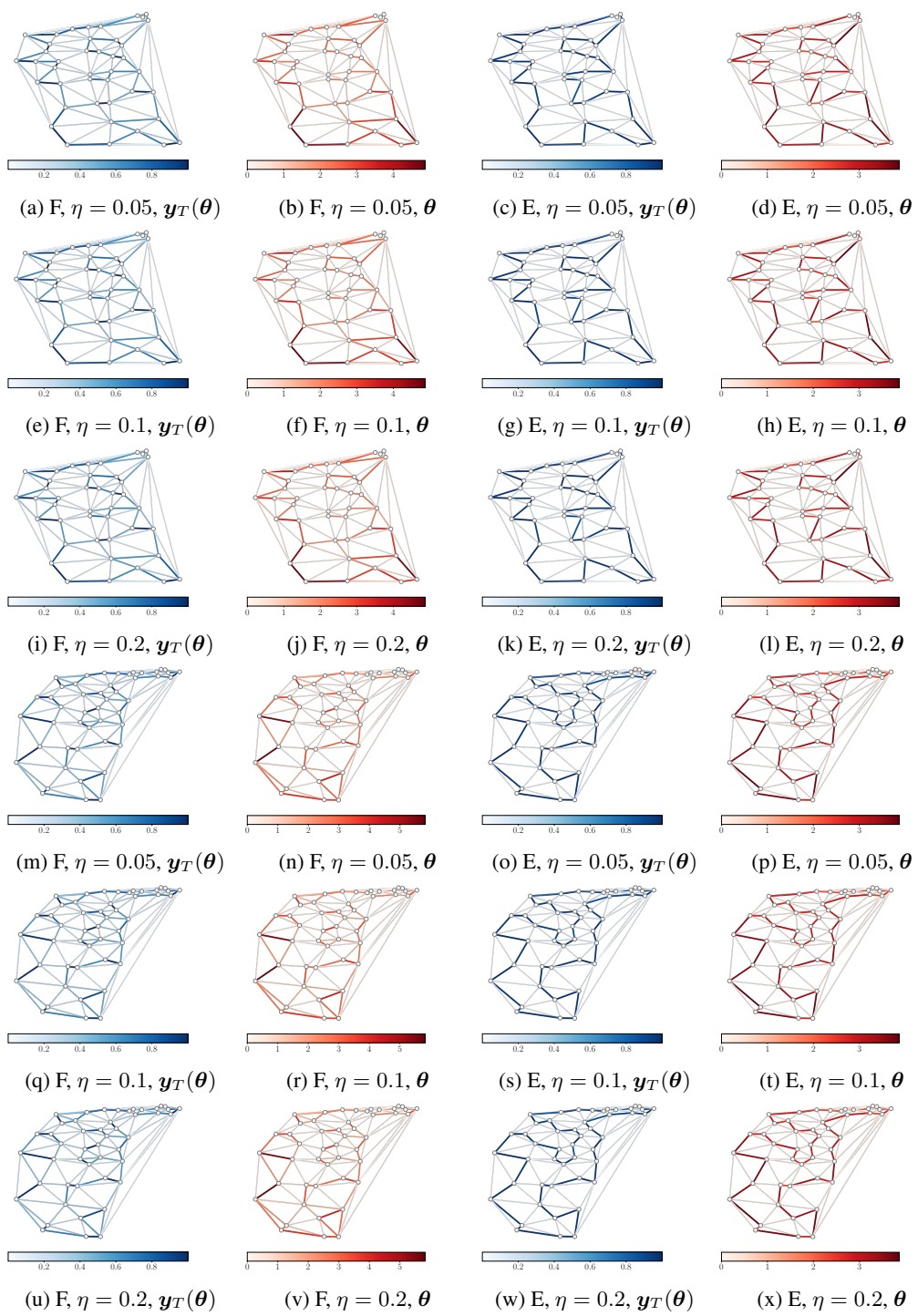

Figure 8: Illustration of DANTZIG (a–l) and ATT (m–x) networks. Blue and red edges represent $\boldsymbol{y}_T(\boldsymbol{\theta})$ and $\boldsymbol{\theta}$ values computed by our method with $T = 300$ and $\eta = 0.05, 0.1, 0.2$ for fractional (F) and exponential (E) cost instances.

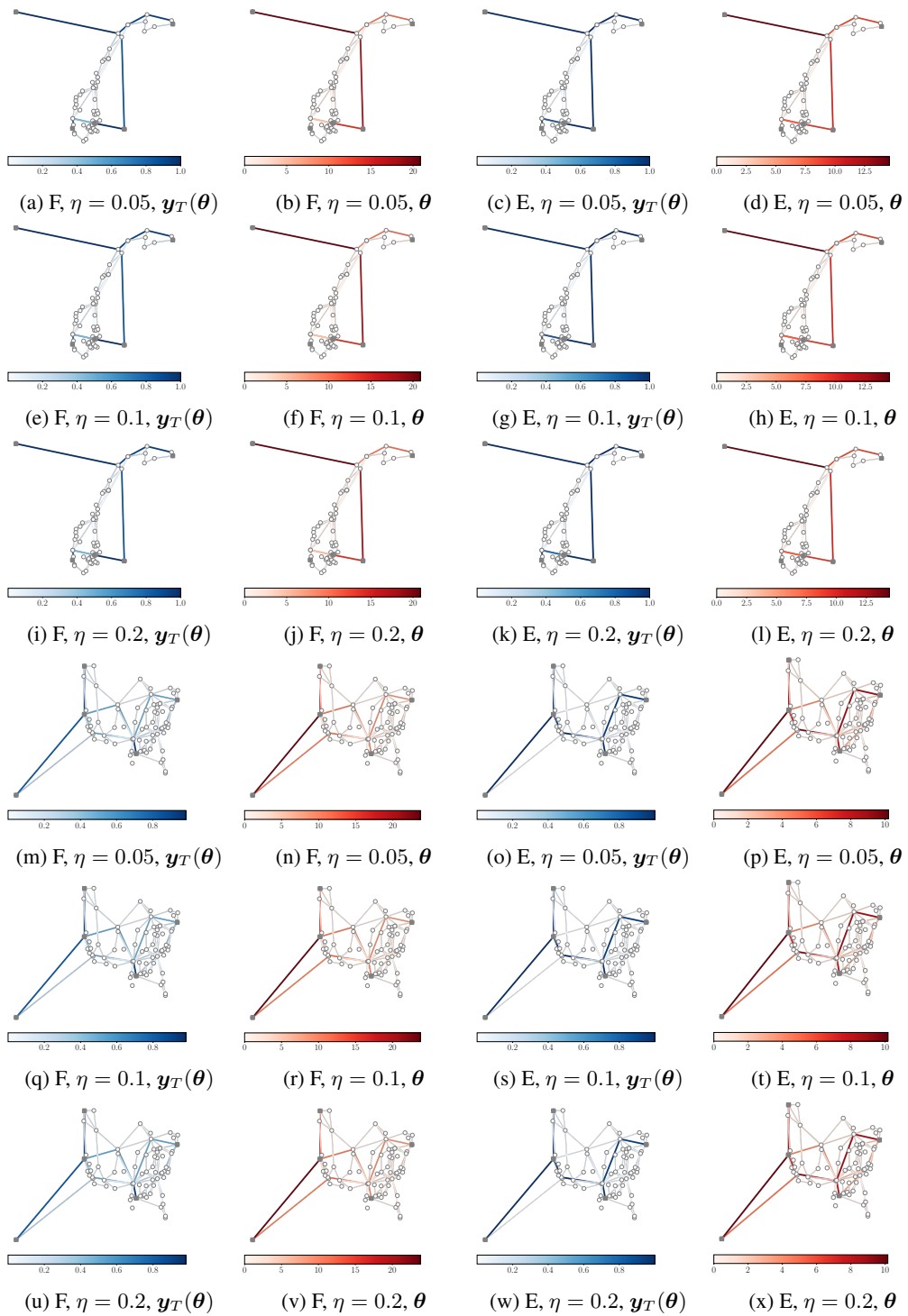

Figure 9: Illustration of UNINETT (a–l) and TW (m–x) networks. Five square vertices in each network are terminals. Blue and red edges represent $\boldsymbol{y}_T(\boldsymbol{\theta})$ and $\boldsymbol{\theta}$ values computed by our method with $T = 300$ and $\eta = 0.05, 0.1, 0.2$ for fractional (F) and exponential (E) cost instances.