# OpenReview forum: "Differentiable Equilibrium Computation with Decision Diagrams for Stackelberg Models of Combinatorial Congestion Games"
_NeurIPS.cc/2021/Conference — NeurIPS 2021 Poster_

### Official Review · Reviewer_WtUV · 2021-07-16

**Rating:** 6
**Confidence:** 3

**Summary:**

This paper studies the problem of designing parameters of combinatorial congestion games, where selfish non-atomic agents choose an optimal route/network to optimize their own objectives. The combinatorial congestion games are a special type of potential games. The equilibrium of a congestion game with given parameters can be solved by optimizing the potential function. The problem of optimizing the game parameters is therefore a bilevel optimization problem with an equilibrium (potential function optimization problem) involved in the inner problem. Benefited from the recent development of differentiable optimization, the authors propose to solve the bilevel optimization problem by backpropagating the gradient from the argmax of the inner problem to the outer objective to run gradient descent. This end-to-end gradient descent approach can avoid solving the bilevel optimization problem with complex constraints by directly running end-to-end gradient descent.

The challenge involved in this end-to-end differentiable optimization approach is how to efficiently compute the gradient of a complex optimization problem (equilibrium computation/potential function optimization) with high-dimensional constraints. The authors leverage the Frank-Wolfe algorithm to solve the optimization problem and to get the corresponding gradient of the optimal solution. The use of the Frank-Wolfe algorithm decomposes the optimization problem into an iterative algorithm composed of multiple minimization problems, where each of them is relaxed to a softmin in order to make them differentiable. To solve the softmin problem with complex constraints, they use zero-suppressed binary decision diagrams (ZDD) to compactly represent the decision variables, which can speed up the optimization step while keeping the differentiability. The authors also utilize Nesterov's acceleration to improve the convergence of the Frank-Wolfe algorithm to $O(1/T^2)$, with the differentiability maintained.

**Limitations And Societal Impact:**

The authors adequately address negative societal impact and limitations. The main limitations are the structure of the potential games and objectives. The same approach does not work when the games are not potential games. The theoretical guarantee of Frank-Wolfe style algorithms also fails when the objectives are not convex. Lastly, the algorithm largely relies on the use of zero-suppressed binary decision diagrams due to the complex constraints. The first two limitations are problem-specific and thus less a concern to me. The last limitation is discussed in the paper that ZDD can be precomputed to reduce its expensive computation cost.


**Main Review:**

The contribution of the paper falls mostly into the claim of making Nesterov’s acceleration differentiable. From the perspective of differentiable optimization, due to the complex constraints in this problem, the authors use the iterative Frank-Wolfe algorithm to optimize the potential function. Thus the optimization operators involved in the algorithm are simply argmin/argmax only. To my knowledge, there is no need to use KKT conditions and they also explained in the paper that they can’t efficiently express the complex KKT conditions as well. Thus the technique used in the problem is more about relaxing argmin to softmin in the Frank-Wolfe algorithm and Nesterov’s acceleration, and less about generalizing differentiable optimization. Overall, I still consider making Nesterov’s acceleration algorithm differentiable a nice contribution in this paper.

I am familiar with differentiable optimization but less familiar with the recent literature on Nesterov’s acceleration. I agree that the main contribution is more on Nesterov's acceleration part. Making Nesterov’s acceleration differentiable is new to me from the perspective of differentiable optimization, but it is relatively a smaller modification compared to previous works in differentiable optimization using Frank-Wolfe algorithms, e.g., Wilder et al. [59] used KKT conditions as opposed to unrolling the entire Frank-Wolfe algorithm to achieve differentiability. I would reserve my judgment on Nesterov’s part to other reviewers, and I would appreciate it if the authors could elaborate more on some similar literature of relaxing argmin/argmax in Nesterov’s acceleration.

Post-response: I am satisfied with the explanation and the additional elaboration of Nesterov’s acceleration that the authors gave. I agree that KTT conditions can sometimes be unnecessarily high-dimensional and complicated. It makes sense to me to sometimes consider unrolling and relaxing argmin/argmax instead of differentiating through the KKT conditions.

**Time Spent Reviewing:**

8

---

> ### Author Response · Authors · 2021-08-10
> **Our Response to Reviewer WtUV**
>
> We are sincerely grateful to the reviewer for providing valuable comments based on deep knowledge of differentiable optimization.
>
> > *Making Nesterov's acceleration differentiable is new to me from the perspective of differentiable optimization, but it is relatively a smaller modification compared to previous works in differentiable optimization using Frank-Wolfe algorithms, e.g., Wilder et al. [59] used KKT conditions as opposed to unrolling the entire Frank-Wolfe algorithm to achieve differentiability. [...] I would appreciate it if the authors could elaborate more on some similar literature of relaxing argmin/argmax in Nesterov's acceleration.*
>
> Nesterov's acceleration is thought to be a somewhat mysterious technique, as mentioned in, e.g., [58] and [Wibisono et al. 2016]. Researchers, however, seem to think that the use of regularized argmin/argmax is crucial for inducing the acceleration, and thus regularized argmin/argmax appears in most studies. For example, a seminal work by Allen-Zhu and Orecchia [3] elucidates how regularized argmin (more precisely, a mirror descent step) yields the acceleration, and their result is also intelligibly described in a recent book [Vishnoi 2021, Chapter 8].
>
> In the context of Nesterov's acceleration, however, regularized argmin/argmax has been utilized **only for acceleration**. To the best of our knowledge, our work is the first to interpret such regularized argmin as relaxation (or smoothing) of argmin, which is a key idea to obtaining the **differentiable and accelerated** optimization algorithm.
>
> Given the background on Nesterov's acceleration, we would like to highlight the technical novelty of our idea, which can be broken down as follows:
>
> 1. It has been known that regularization plays a crucial role in inducing Nesterov's acceleration. We utilize this as a differentiable relaxation of argmin. Specifically, using the KL divergence as a regularizer, we write the regularized argmin by differentiable softmin, as detailed in Appendix B.
>
> 2. To efficiently compute the softmin over a massive number of combinatorial strategies, we employ weight pushing on zero-suppressed binary decision diagrams (ZDDs). Importantly, this uses only elementary arithmetic operations as in Algorithm 2 (and mentioned in ll. 220--224). Thus, our algorithm can be implemented so as to accept reverse-mode automatic differentiation.
>
> That is, our algorithm is a careful combination of Nesterov's acceleration and ZDD-based weight pushing, which enables efficient equilibrium computation while keeping the differentiability even in the presence of complicated combinatorial strategies.
>
> We certainly agree with the reviewer that the recent advances in implicit differentiation, e.g., Wilder et al. [59] and Agrawal et al. [2], are of great importance. On the other hand, considering that equation systems of the KKT condition are sometimes prohibitively large under complicated combinatorial structures, we believe that our method is also of considerable significance. We would appreciate it if the reviewer could consider raising the score by taking the above technical significance, together with the potential social benefit of our method, into account.
>
> #### **References**
> - A. Wibisono, A. C. Wilson, and M. I. Jordan. A variational perspective on accelerated methods in optimization. *Proceedings of the National Academy of Sciences*, 113(47):E7351--E7358, 2016.
> - N. K. Vishnoi. *Algorithms for Convex Optimization*. Cambridge University Press, 2021.

---

### Official Review · Reviewer_2skQ · 2021-07-16

**Rating:** 7
**Confidence:** 4

**Summary:**

The authors of this paper aim to find a combinatorial congestion game that optimizes a given objective over some parameterized class of games, assuming that the players in the game attain a Wardrop equilibrium. The problem is formulated using a framework of Stackelberg bi-level optimization models, where the designer (leader) chooses the game's parameters, and the followers then play the induced game. The authors propose to apply gradient-based methods to approximate the solution. To this end, it remains critical to be able to compute the followers' equilibrium and estimate its gradient, i.e., the changes in the equilibrium when the parameters of the game shift. For this purpose, the authors introduce a differentiable variant of the Frank–Wolfe algorithm. Instead of the strict minimization included in the original version, the authors suggest using softmin. Computing the softmin may be computationally demanding because it sums over all possible strategies, which is (in general) a set exponential in the problem size. More efficient computation can be achieved using the so-called zero-suppressed binary decision diagrams (ZDDs), a data structure enabling calculation of the softmin in time linear in its size. Because the size of the ZDDs is often many orders of magnitude smaller than the cardinality of the set of strategies, ZDDs offer a significant speedup in practice. In the last part of the manuscript, the authors present empirical results achieved with their algorithm and its variants on two classes of games per each strategy representation (Hamiltonian cycles and Steiner trees). The results show that the algorithm can converge to Wardrop equilibria faster than baselines, and the gradient descent running on top of it consequently designs better congestion games than the heuristic baselines.


**Ethical Concerns:**

I do not have any ethical concerns.

**Limitations And Societal Impact:**

I find the limitations the authors mention in Section 5 honest and sufficient. Some discussion of the optimality of the gradient descent might be included for completeness.

**Main Review:**

The paper is well written, all concepts are rigorously explained, and the notation is clear. I appreciate the examples described in the introduction that well illustrate the motivation of this paper. Also, the authors explain well how their work fits into the existing literature on the topic. As far as I can tell, the results are sound, the analyses and the proof of Theorem 1 seem correct to me.

The contributions of this paper may not be as strong since the approach is relatively similar to the method introduced in [42], i.e., a Frank–Wolfe-type algorithm accelerated using the ZDDs. Despite some similarities, this work makes the cut in my eyes because of the introduction of the Stackelberg setting, which required adapting the Frank–Wolfe algorithm for computing the gradients.

While Theorem 1 establishes the convergence guarantee of solutions returned by the Frank–Wolfe-type algorithm to the Wardrop equilibrium, the work does not offer any results on the optimality of the final parameters computed by the gradient descent. If I interpret it correctly, there is no guarantee of how close to the global optimum the computed parameters are, given that the criterion F is non-convex in general. Did the authors try to perform some experiments on very small settings when the globally optimal Stackelberg parameters can be found using, e.g., linearization into MILPs, to see how well the gradient descent can approximate the global solution in practice?

One minor note: the legends of all graphs in Figure 2 obstruct the view of the plots. Since the legend is shared across all the graphs, it might be better to put one legend underneath them all. Similarly also in Figure 3.

### After rebuttal

I would like to thank the authors for a very detailed response to my questions and the amount of effort they put into coming with examples and running the additional experiments. I especially appreciate the example illustrating the difficulty of global optimization in this bilevel setting. My impression of this work remains unchanged, I believe it is a good research that deserves publication.


**Time Spent Reviewing:**

25

---

> ### Author Response · Authors · 2021-08-10
> **Our Response to Reviewer 2skQ**
>
> We extend our sincere gratitude to the reviewer for spending considerable time carefully reading our paper and writing many valuable comments.
>
> > *If I interpret it correctly, there is no guarantee of how close to the global optimum the computed parameters are, given that the criterion F is non-convex in general.*
>
> Certainly, our work does not guarantee the global optimality of computed parameter $\boldsymbol{\theta}$. We will clearly mention this limitation in Section 5.
>
> We would like to illustrate that guaranteeing the optimality of $\boldsymbol{\theta}$ is not easy **even if $F(\boldsymbol{\theta}, \boldsymbol{y})$ is convex in $\boldsymbol{\theta}$ and $\boldsymbol{y}$**. To see this, let us consider
> a simple example with two strategies $\\{1\\}$ and $\\{2\\}$, each of which has cost $\theta_1 y_1$ and $\theta_2 y_2$, respectively. The Stackelberg model for minimizing social cost $F(\boldsymbol{\theta}, \boldsymbol{y})$ under constraints $\theta_1 \ge 0$, $\theta_2\ge0$, and  $\theta_1 + \theta_2 = 1$ can be written as follows:
>
> $$
> \def\minimize{\mathop{\mathrm{minimize}}\limits}
> \def\argmin{\mathop{\mathrm{argmin}}\limits}
> %
> \minimize_{\theta_1, \theta_2 \ge 0, \theta_1 + \theta_2 = 1} \quad
> F(\boldsymbol{\theta}, \boldsymbol{y}) = \theta_1 y_1^2 + \theta_2 y_2^2 \qquad
> \mathrm{subject\ to} \quad
> \boldsymbol{y} = \argmin_{y_1, y_2 \ge 0, y_1 + y_2 = 1}  \frac{\theta_1}{2} y_1^2 + \frac{\theta_2}{2} y_2^2.
> $$
>
> In this case, $F(\boldsymbol{\theta}, \boldsymbol{y})$ is convex in $\boldsymbol{\theta} = (\theta_1, \theta_2)$ and $\boldsymbol{y} = (y_1, y_2)$. However, from $\theta_1 + \theta_2 = 1$, we can confirm that $(y_1, y_2) = (\theta_2, \theta_1)$ is a unique solution to the argmin constraint. Therefore, the above problem can be rewritten as
>
> $$
> \def\minimize{\mathop{\mathrm{minimize}}\limits}
> %
> \minimize_{\theta_1, \theta_2 \ge 0, \theta_1 + \theta_2 = 1} \quad
> \theta_1 \theta_2^2 + \theta_2 \theta_1^2 = \theta_1 \theta_2,
> $$
>
> where the objective function, $\theta_1\theta_2$, is non-convex in $\boldsymbol{\theta}$. This example suggests that the main difficulty comes from the bilevel structure of the problem, i.e., the argmin constraint can easily make the leader's optimization problem non-convex.
>
> Considering the above, how to guarantee the optimality of $\boldsymbol{\theta}$ seems to be an intrinsically hard question, which should be addressed in more theoretical studies on simpler Stackelberg games. By contrast, our purpose is to address practical Stackelberg models with complicated combinatorial strategies. Therefore, we focused on computing gradients $\nabla F(\boldsymbol{\theta}, \boldsymbol{y}(\boldsymbol{\theta}))$ rather than analyzing the optimality of computed $\boldsymbol{\theta}$.
>
> > *Did the authors try to perform some experiments on very small settings when the globally optimal Stackelberg parameters can be found using, e.g., linearization into MILPs, to see how well the gradient descent can approximate the global solution in practice?*
>
> We appreciate the reviewer's helpful suggestion. Having read the comment, we performed additional experiments on small instances to check whether computed $\boldsymbol{\theta}$ is close to the global optimum or not. We confirmed that the projected gradient descent (used in Section 4.2) successfully found optimal $\boldsymbol{\theta}$, although the optimality is not theoretically guaranteed as mentioned above. We will add the results to the appendix. Below are the details of the experiments.
>
>
> ### **Experiments on empirical convergence to global optimum**
> Let $G = (V, E)$ be an undirected graph such that $V = \\{ s, u, v, t \\}$ and $E = \\{ (s, u), (s, v), (u, v), (u, t), (v, t) \\}$, where the edges are numbered from $1$ to $5$ in this order. The graph looks like as follows:
>
> ```
>     s
>    / \
>   1   2
>  /     \
> u---3---v
>  \     /
>   4   5
>    \ /
>     t
> ```
>
> We let strategy set $\mathcal{S}$ be the set of all simple $s$-$t$ paths. As with the problem setting in Section 4, we associate fractional or exponential cost $c_i(y_i; \boldsymbol{\boldsymbol{\theta})}$ with each edge in $E$, and we impose the following constraints on $\boldsymbol{\theta} \in\mathbb{R}^E$: $\boldsymbol{\theta} \ge 0$ and $\boldsymbol{\theta}^\top \boldsymbol{1} = |E| = 5$. We aim to minimize social cost $F(\boldsymbol{\theta}, \boldsymbol{y}(\boldsymbol{\theta}))$ by optimizing $\boldsymbol{\theta}$ values.
>
> As in Section 4.2, we computed equilibrium $\boldsymbol{y}(\boldsymbol{\theta})$ using Algorithm 1 with $\eta = 0.1$ and $T = 300$, and performed the projected gradient descent to minimize $F(\boldsymbol{\theta}, \boldsymbol{y}(\boldsymbol{\theta}))$, where the gradient $\nabla F(\boldsymbol{\theta}, \boldsymbol{y}(\boldsymbol{\theta}))$ was computed by backpropagation.
>
> To obtain (approximations of) globally optimal $\boldsymbol{\theta}$, we performed an exhaustive search over the feasible region, where the step size was set to $0.05$ (we did not use MILP solvers since they did not seem to be very useful for bilevel optimization).
>
> #### **Results on fractional costs**
> A global optimal solution found by the exhaustive search was $\boldsymbol{\theta} = (0, 2.5, 0, 0, 2.5)$, whose social cost $F(\boldsymbol{\theta}, \boldsymbol{y}(\boldsymbol{\theta}))$ was $6.444$.
>
> Our method started from $\boldsymbol{\theta} = \boldsymbol{1}$, whose social cost was $7.000$, and returned $\boldsymbol{\theta} = (1.25, 1.25, 0, 1.25, 1.25)$ with social cost $6.444$. Although the solution is different from that of the exhaustive search, both attain the identical social cost. Thus, the solution of our method is also globally optimal.
>
> #### **Results on exponential costs**
> A global optimal solution found by the exhaustive search was $\boldsymbol{\theta} = (0, 2.5, 0, 0, 2.5)$ with social cost $3.517$.
>
> Our method started from $\boldsymbol{\theta} = \boldsymbol{1}$ with social cost $5.678$ and reached $\boldsymbol{\theta} = (0, 2.5, 0, 0, 2.5)$ with social cost $3.517$. Along the way, our method was about to be trapped in $\boldsymbol{\theta} = (1.25, 1.25, 0, 1.25, 1.25)$ with social cost $4.865$, which seems to be a saddle point. However, it successfully got out of there and returned the global optimal solution.
>
> #### **Note on computation time**
> In both settings, our method reached the global optimum in less than $30$ iterations of the projected gradient descent, which took less than $20$ ms. Meanwhile, the exhaustive search took about $1000$ seconds.

---

### Official Review · Reviewer_3ev8 · 2021-07-23

**Rating:** 7
**Confidence:** 3

**Summary:**

This paper focus on the Stackelberg model of combinatorial congestion games (CCGs), which is important for the case involved the leader and multiple self-interested followers. The main contribution of this paper is proposing a fully differentiable framework which can take the advantage of automatic differentiation.

My main concern is the contribution of this paper. There are two key techiniques in the proposed algorithms, softmin and Zero-suppressed binary decision diagrams (ZDD), where softmin is a widely used technique to make argmax operation to be differetiable and the ZDD technique is proposed in the 2020 AAAI paper, Practical Frank–Wolfe Method with Decision Diagramsfor Computing Wardrop Equilibrium of Combinatorial Congestion Games. I would like to ask authors to clearly state the difference of the techniques in this paper and the one in the 2020 AAAI paper.

Minor issues and typos:

"practical utility" in line 60, "utility" -> usage or application?
"Our method introduces benefits by improving the designs of social infrastructures" is strange.

**Ethical Concerns:**

No.

**Limitations And Societal Impact:**

Yes.

**Main Review:**

Originality
A recent paper [42] proposed ZDDs method to tackle the combinatorial nature of CCGs. This paper moves one step forward to consider the Stackelberg models.

Quality
The submission is technically sound.

Clarity
Clearly written.



Significance
The studied problem is not new, and the proposed method are mainly based on [42]. However, the Stackelberg model is very important, and the proposed technical novelties are very effective, thus I think the paper can be accepted.


**Time Spent Reviewing:**

7

---

> ### Author Response · Authors · 2021-08-10
> **Our Response to Reviewer 3ev8**
>
> We express our sincere thanks to the reviewer for careful reading and positive feedback.
>
> > *The studied problem is not new, and the proposed method are mainly based on [42]. However, the Stackelberg model is very important, and the proposed technical novelties are very effective, thus I think the paper can be accepted.*
>
> We are pleased to see that the reviewer has found our work to be important and novel. As mentioned in the comment, although our work is inspired by Nakamura et al. [42], our method has technical novelty. Below we would like to summarize essential points of our work to highlight the novelty and significance.
>
> 1. We utilize Nesterov's acceleration to obtain a differentiable optimization algorithm (Algorithm 1), whose accelerated convergence is guaranteed by Theorem 1. This idea is of particular interest in the context of differentiable optimization since we are the first to explicitly show that Nesterov's acceleration can be leveraged for obtaining **differentiable and accelerated** optimization algorithms (for further details, please see the response to [Reviewer WtUV](https://openreview.net/forum?id=umuW_b77q9A&noteId=MjO6pdbCwHT)).
>
> 2. We employ ZDD-based softmin (Algorithm 2) for dealing with a vast number of combinatorial strategies without breaking the differentiability of computed equilibrium $\boldsymbol{y}(\boldsymbol{\theta})$. **This technique, called weight pushing, is an essential difference from ZDD-based argmin of [42], which breaks the differentiability.**
>
> We would appreciate it if the reviewer could consider raising the score by taking the above technical novelty and significance, together with the potential benefit our work can bring in designing social infrastructures, into account.

---

### Official Review · Reviewer_Xufg · 2021-07-23

**Rating:** 6
**Confidence:** 2

**Summary:**

The problem of computing equilibria in CGs is consider an easy task whenever the strategy space has a compact representation, since it turns out that the users minimize a convex function. When having to decide on some parameters of the CG in order to optimize the performance at equilibrium we get a bi-level optimization problem, where in the upper level the designer sets the parameters and in the lower level the users optimize their function reaching an equilibrium. Solving this bi-level problem is regarded to be far more difficult. This work deals with the latter problem and combines optimization techniques to give an algorithm that performs well in practice.

**Limitations And Societal Impact:**

yes

**Main Review:**

The problem of computing equilibria in CGs is consider an easy task whenever the strategy space has a compact representation, since it turns out that the users minimize a convex function. When having to decide on some parameters of the CG in order to optimize the performance at equilibrium we get a bi-level optimization problem, where in the upper level the designer sets the parameters and in the lower level the users optimize their function reaching an equilibrium. Solving this bi-level problem is regarded to be far more difficult. This work deals with the latter problem and combines optimization techniques to give an algorithm that performs well in practice.

The goal of the network designer is to minimize a function F that depends on the choice of the strategies of the players (let it be described by vector y) and a parameter vector \theta. The choice of \theta is on the designer and users reach an equilibrium based on that choice, minimizing a different function f so that y=argmin f.  For the case of fixed \theta we get a single level program computing an equilibrium of the CG. A recent work (Nakamura et al. 20) used a data structure (zero-suppressed binary decision diagrams (ZDDs)) to get a compact strategy space representation for interesting cases and applied a Frank-Wolfe style algorithm that performs well whenever the representation using ZDDs is indeed compact. Here, for the bi-level program, this technique does not work as is because of non differentiability wrt \theta. To get differentiability, the authors use softmin instead of argmin in the computation of y, while keeping the ZDDs representation for the strategy space, and provide an algorithm that seemingly performs well in practice.

Pros. The idea of using softmin (and making it work) in order to make the equilibrium differentiable wrt \theta is interesting

Cons. Limited theoretical guarantees (although, in general this is difficult for bi-level programs)

If we fix \theta, how does Algorithm 1 compare to the algorithm of Nakamura et al.?

**Time Spent Reviewing:**

7

---

> ### Author Response · Authors · 2021-08-10
> **Our Response to Reviewer Xufg**
>
> We greatly appreciate the reviewer's careful reading and insightful comments.
>
> > *Limited theoretical guarantees (although, in general this is difficult for bi-level programs)*
>
> Certainly, our work is limited regarding the theoretical guarantee on computed parameter $\boldsymbol{\theta}$. We thank the reviewer for pointing this out. We will clearly mention this in Section 5.
>
> As mentioned in the review comment, guaranteeing the optimality of $\boldsymbol{\theta}$ is not easy, even in simple Stackelberg models (for example, please see the response to [Reviewer 2skQ](https://openreview.net/forum?id=umuW_b77q9A&noteId=948bR7J-H_O)). Since our primary purpose is to develop an efficient gradient computation method for addressing practical Stackelberg models with complicated combinatorial strategies, we have left the theoretical analysis of the optimality of $\boldsymbol{\theta}$ for future work. In the response to [Reviewer 2skQ](https://openreview.net/forum?id=umuW_b77q9A&noteId=948bR7J-H_O), we present additional experiments to confirm that our method can empirically find optimal $\boldsymbol{\theta}$ for small instances.
>
> Also, we would like to mention that our work provides a meaningful theoretical result regarding the convergence of equilibrium computation (Theorem 1). This result is of particular interest in the context of differentiable optimization, as also mentioned by [Reviewer WtUV](https://openreview.net/forum?id=umuW_b77q9A&noteId=MjO6pdbCwHT), since it is the first to explicitly show that Nesterov's acceleration can be utilized to obtain **differentiable and accelerated** optimization algorithms. We would appreciate it if the reviewer could consider raising the score taking this contribution to the area of differentiable optimization into account.
>
> > *If we fix $\boldsymbol{\theta}$, how does Algorithm 1 compare to the algorithm of Nakamura et al.?*
>
> Generally speaking, the algorithm of Nakamura et al. [42] can run faster for fixed $\boldsymbol{\theta}$ (but its output is non-differentiable in $\boldsymbol{\theta}$ unlike ours). This is because their algorithm is based on a fast algorithm called the fully corrective Frank--Wolfe [33], which empirically converges very quickly and achieves a linear convergence rate under the assumption that objective functions are strongly convex (not strictly convex). We will clearly mention this in Section 5.
>
> To the best of our knowledge, no existing linearly convergent Frank--Wolfe variants preserve the differentiability in $\boldsymbol{\theta}$. For example, those in [33] use non-differentiable argmin. To develop differentiable algorithms that can achieve linear convergence rates as with [33, 42] would be interesting future work, as mentioned in ll. 346--347.

---

### Decision · Program_Chairs · 2021-09-27

**Decision:**

Accept (Poster)

**Comment:**

I and the Reviewers have positive opinions on the paper and no major concern has been raised during the reviewing and discussion phases. Furthermore, the Reviewers found the rebuttals very useful, illustrating and clarifying better several points. In particular, I encourage the authors to add the example provided in the rebuttal to Reviewer 2skQ in the camera ready to their paper (the authors could add the example in the supplementary material if no space is available in the main body of the paper).